# Diffusion-Denoised Hyperspectral Gaussian Splatting

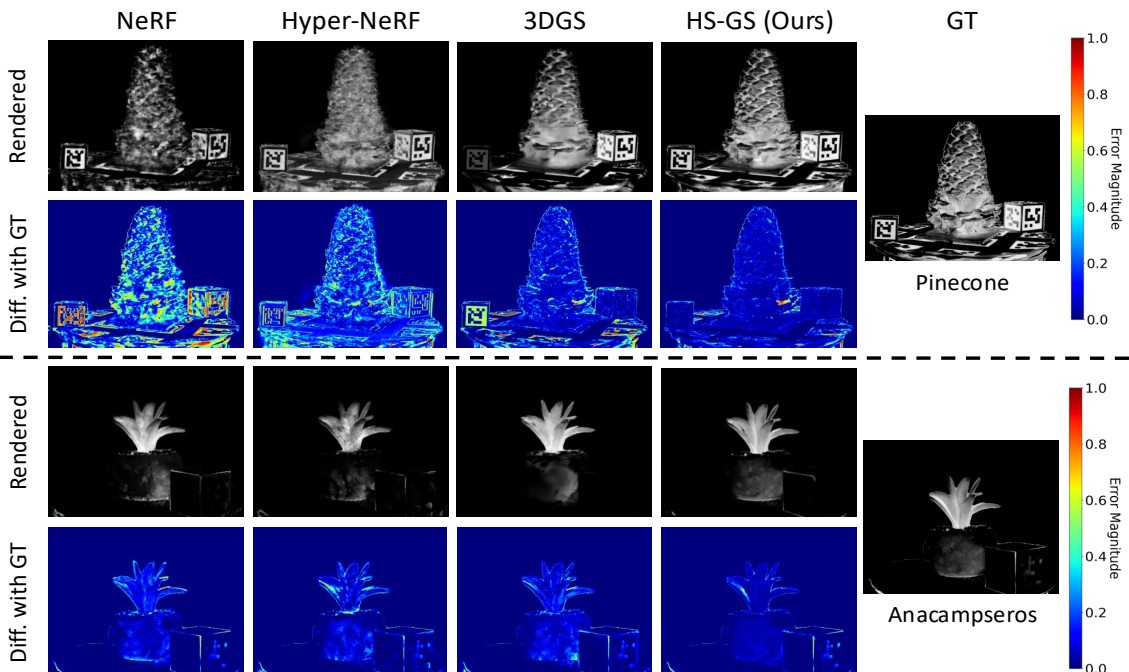

Figure 1: We propose a Diffusion-Denoised Hyperspectral Gaussian Splatting (HS-GS) for reconstructing agricultural scenes and enabling novel view synthesis under hyperspectral imaging. Compared with NeRF (Mildenhall et al., 2020), Hyper-NeRF (Chen et al., 2024a) and 3DGS (Kerbl et al., 2023), ours can render high-quality images with fine-grained spectral details, and significantly reduce reconstruction errors.

## Abstract

Hyperspectral imaging (HSI) has been widely used in agricultural applications for non-destructive estimation of plant nutrient composition and precise determination of nutritional elements of samples. Recently, 3D reconstruction methods have been used to create implicit neural representations of HSI scenes, which can help localize the target object's nutrient composition spatially and spectrally. Neural Radiance Field (NeRF) is a cutting-edge implicit representation that can be used to render hyperspectral channel compositions of each spatial location from any viewing direction. However, it faces limitations in training time and rendering speed. In this paper, we propose Diffusion-Denoised Hyperspectral Gaussian Splatting (HS-GS), which combines the state-of-the-art 3D Gaussian Splatting (3DGS) with a diffusion model to enable 3D explicit reconstruction of the hyperspectral scenes and novel view synthesis for the entire spectral range. To enhance the model's ability to capture fine-grained reflectance variations across the light spectrum and leverage correlations between adjacent wavelengths for denoising, we introduce a wavelength encoder to generate wavelength-specific spherical harmonics offsets. We also introduce a novel Kullback–Leibler divergence-based loss to mitigate the spectral distribution gap between the rendered image and the ground truth. A diffusion model is further applied for denoising the rendered images

and generating photorealistic hyperspectral images. We present extensive evaluations on five diverse hyperspectral scenes from the Hyper-NeRF dataset to show the effectiveness of our HS-GS. The results demonstrate HS-GS has achieved the new state-of-the-art performance among all the previously published methods. Code will be released soon.

# 1  Introduction

Human eyes can respond to wavelengths within the visible spectrum, approximately 380 nm to 750 nm. To match human vision, conventional RGB cameras capture visible light in three broad bands: red, green, and blue. In contrast, hyperspectral cameras capture wavelengths across a broader spectrum beyond the visible light, allowing the analysis of how objects interact with many more bands of light. For a hyperspectral image, each pixel consists of $N$ bands/channels with $N$ being dependent on the range of wavelengths captured by the sensor and the spectral resolution of the sensor. Since different materials exhibit distinct reflectance rates at various wavelengths, they can be identified through their emission spectral signatures (Liang et al., 2008). This capability has made hyperspectral imaging (HSI) a valuable tool in various applications, including agriculture, food quality control, construction, and environmental monitoring. In agriculture, HSI has been utilized for non-destructive nutrient analysis for crops, and precise determination of the material composition of a plant sample (Corti et al., 2017; Liu et al., 2021). Furthermore, having a 3D spatial reconstruction of hyperspectral images can provide even more detailed analysis of such material subjects, which is critical for localizing the presence of different materials in the 3D physical world. This approach lays the groundwork for building digital twins—virtual replicas of physical objects or environments that include not only spatial geometry but also functional and material properties. By embedding real-world data, 3D digital twins can simulate interactions, predict behaviors, and provide immersive and data-driven insights.

The development of digital twins has paved the way for constructing detailed representations of real-world environments, offering capabilities like mineral composition analysis and crop yield estimation. The integration of spectral information in an agricultural digital twin holds immense potential for its downstream applications, as demonstrated by the enhanced fruit counting accuracy (Meyer et al., 2024) and fruit detection performance in challenging low-light conditions (Chopra et al., 2024). Towards developing a digital twin for *precise agriculture*, we focus on modeling highly detailed 3D spectral information for crops to facilitate applications such as crop monitoring (Qin et al., 2023), yield estimation (Yang et al., 2021; Li et al., 2021), and non-destructive nutrient assessment (Hu et al., 2021; Bai et al., 2018). These capabilities can enhance resource management and food productivity to address climate change and population escalating challenges, achieve precise agriculture for personalized food production to improve public health, further underscoring the transformative impact of agricultural digital twins.

Neural Radiance Fields (NeRF) (Mildenhall et al., 2020) with implicit neural representation has become prominent for 3D reconstruction of complex scenes. Previous studies (Chen et al., 2024a) have explored the feasibility of extending NeRF to model 3D spectral information using hyperspectral images. However, these methods inherit the limitations from the original NeRF, including slow convergence, long training time, and the tendency to overfit to image noise. This poses a significant challenge to HSI because hyperspectral images are inherently high-dimensional and noisy. As the width of the spectral band decreases, the captured signal is weakened since narrower spectral bands permit less light to reach the sensor (Rasti et al., 2018), making the sensing system more vulnerable to noise. Thus, how to simultaneously *denoise* and *reconstruct* high-quality scenes for HSI data remains an open problem.

Recently, 3D Gaussian Splatting (3DGS) (Kerbl et al., 2023) has been proposed for reconstructing high-quality 3D scenes and novel view synthesis in real time. 3DGS represents the scene as a set of anisotropic 3D Gaussians and renders images with a fast rasterization algorithm. Inspired by 3DGS, we propose a **Diffusion-Denoised Hyperspectral Gaussian Splatting (HS-GS)** framework to extend the vanilla Gaussian Splatting for building a 3D hyperspectral scene representation and subsequently rendering novel-view hyperspectral images. Compared to NeRF-based methods, ours can render noise-free hyperspectral images even with noisy input data.

To summarize, our main contributions are as follows:

- We propose a novel HS-GS framework for 3D reconstruction and novel view synthesis of HSI scenes, extending explicit neural representations to the hyperspectral domain by modeling full spectral radiance directly, rather than through compressed features as in prior work.

- We introduce a **wavelength encoder** to embed wavelength-specific spherical harmonics information into 3D Gaussians. We also design a Kullback-Leibler (KL) divergence-based **spectral loss** that can drive Gaussian reflectances to be aligned with ground truth spectral distributions.

- We integrate a **hyperspectral diffusion model** which de-noises hyperspectral images rendered from 3DGS. The diffusion model improves both spatial and spectral fidelity by learning to correct structural and color artifacts produced in the upstream rendering process. We train the enhanced 3DGS pipeline and the hyperspectral diffusion model together in an end-to-end manner.

- We conduct extensive quantitative and qualitative evaluations on five diverse hyperspectral scenes from the Hyper-NeRF dataset to demonstrate the effectiveness of our HS-GS framework. HS-GS achieves the state-of-the-art performance among all the published methods.

## 2 Related Work

### 2.1 Hyperspectral Imaging

The primary differences between hyperspectral and RGB imaging lie in spectral resolution. While RGB images capture light in three broad bands which are red, green, and blue, hyperspectral images encode fine-grained spectral information across tens to hundreds of narrow and contiguous wavelength bands ranging from ultraviolet (380 nm) to near-infrared (1100 nm) (Ahmad et al., 2024). This provides us with rich spectral information which can be used for detailed material discrimination and agricultural applications such as non-destructive nutrient assessment, pollutant detection, and mineral composition analysis. However, hyperspectral cameras are highly susceptible to noise. Due to the usage of narrow-band filters, a small number of photons reach the sensor at each band, resulting in a lower signal-to-noise ratio. In practical settings, the vulnerability is further aggravated by environmental and lighting conditions (Liang et al., 2013; Zeng et al., 2024; Li et al., 2023; Fu et al., 2015). Some denoising strategies are proposed recently to exploit spectral correlations of adjacent spectral bands which share the same structure (Rasti et al., 2018; Cao et al., 2016) to effectively suppress noise and enhance image quality.

### 2.2 3D Reconstruction for Hyperspectral Imaging

NeRF (Mildenhall et al., 2020) has revolutionized novel view synthesis by implicitly modeling volumetric scenes using Multi-Layer Perceptrons (MLPs). It takes as input a 3D position and viewing direction and outputs color and density values, which are integrated along camera rays via volumetric rendering. X-NeRF (Poggi et al., 2022) models cross-spectral consistency across sensors with different spectral responses. Another approach generalizes the RGB output of NeRF to $N$ channels to capture hyperspectral reflectance (Ma et al., 2024). Hyper-NeRF (Chen et al., 2024a) introduces a wavelength-aware MLP that jointly encodes spatial position, viewing direction, and wavelength information to predict radiance and density for each band. However, these models suffer from slow rendering speed and vulnerability to noise.

3DGS (Kerbl et al., 2023) offers an explicit and point-based representation that enables real-time rendering with high visual quality. The scene is represented as a set of anisotropic Gaussians, each parameterized by its position, scale, orientation, opacity, and spherical harmonics (SH) for view-dependent color. A differentiable rasterizer is designed for efficient optimization and fast rendering. 3DGS has been extended to dynamic scene reconstruction (Yang et al., 2024b; Wu et al., 2024; Luiten et al., 2024). One concurrent work, Hyper-GS (Thirgood et al., 2025), leverages an autoencoder to compress hyperspectral inputs into a latent space and uses an MLP to decode view-dependent radiance. However, it operates 3D Gaussians in a compressed feature space. In contrast, our proposed HS-GS framework models hyperspectral radiance directly, which avoids spectral compression, improves robustness to noise, and preserves full-resolution spectral fidelity, especially in challenging regions where prior methods degrade.

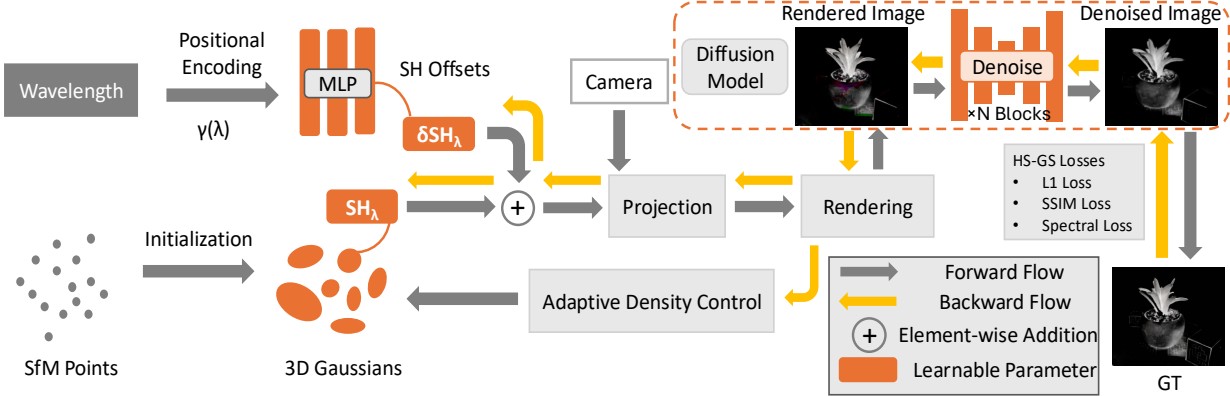

Figure 2: Overview of our HS-GS framework. HS-GS enhances 3DGS with a wavelength encoder that maps positional embeddings of wavelength through an MLP to predict wavelength-dependent SH offsets, a spectral loss aligning predicted and ground truth spectral distributions, and a conditional diffusion module that refines noisy output rendered by 3DGS to improve its spectral and spatial fidelity.

## 2.3 Hyperspectral Diffusion Models

Diffusion models (Ho et al., 2020) are a class of generative models that learn to reverse a gradual noising process to generate high-fidelity samples from noise. Recently, they have been applied to hyperspectral image enhancement and super-resolution tasks (Wang et al., 2024; Cao et al., 2024; Cheng et al., 2025). These methods treat the clean hyperspectral image as the result of a reverse diffusion process, starting from noise and progressively recovering both spectral and spatial details. Wang et al. (2024) propose a group-wise autoencoder combined with a diffusion model for the spectral super-resolution task, while Cao et al. (2024) introduce disentangled modulation strategies to preserve spatial and spectral characteristics during image sharpening. However, these approaches operate purely in the 2D image domain and lack explicit modeling of 3D scene geometry and view-dependent effects. As a result, they are unsuitable for novel view synthesis or spatially consistent hyperspectral scene reconstruction.

To overcome these limitations, we integrate a conditional diffusion model directly into the reconstruction and rendering pipeline based on 3DGS (Kerbl et al., 2023). The rendered hyperspectral images that exhibit geometry and color artifacts are further denoised and refined with a hyperspectral diffusion model. Prior works like GaussianObject (Yang et al., 2024a) and MVSplat360 (Chen et al., 2024b) use frozen Stable Diffusion models trained on large-scale RGB datasets to enhance natural image rendering. In contrast, due to the scarcity and sensor variability of hyperspectral data, our method trains a hyperspectral diffusion model with 3D Gaussian Splatting jointly, incorporating spectral characteristics into diffusion process for more accurate hyperspectral scene reconstruction.

## 3   Method

The overall framework of Diffusion-Denoised Hyperspectral Gaussian Splatting (HS-GS) is illustrated in Figure 2. Our method builds upon 3D Gaussian Splatting (Kerbl et al., 2023) by introducing wavelength-aware modules that enable high-fidelity hyperspectral rendering. We initialize a set of 3D Gaussians from multi-view hyperspectral images with known camera poses, and extend the 3DGS pipeline with the following three key components. First, we introduce a wavelength encoder that maps each input channel's wavelength to a spectral offset through a positional encoding followed by an MLP. These offsets are applied to the spherical harmonics coefficients of the Gaussians to model wavelength-dependent appearance. Secondly, we incorporate a spectral loss that enforces alignment between predicted and ground truth spectral distributions at pixel-wise level. This loss combines Kullback-Leibler divergence and cosine similarity to promote both distributional and angular spectral consistency. Finally, we include a conditional diffusion model that refines the noisy images rendered by 3DGS. This module learns to denoise the output image conditioned on both spatial and spectral context, improving fine structure and reducing residual artifacts. Together, these

components allow HS-GS to accurately synthesize spatially-coherent and spectrally-aligned hyperspectral images from sparse multi-view input, especially for noisy spectral bands.

## 3.1 Hyperspectral Gaussian Splatting

### 3.1.1 3D Gaussian Splatting

3DGS (Kerbl et al., 2023) represents a scene using a set of 3D Gaussians, where each Gaussian is represented by its mean position $\boldsymbol{\mu}$ and covariance matrix $\boldsymbol{\Sigma}$:

$$G(\mathbf{x}) = e^{-\frac{1}{2}(\mathbf{x}-\boldsymbol{\mu})^T \boldsymbol{\Sigma}^{-1}(\mathbf{x}-\boldsymbol{\mu})}, \tag{1}$$

where $\boldsymbol{\Sigma}$ can be further decomposed into a rotation matrix $\boldsymbol{R}$ and a scaling matrix $\boldsymbol{S}$:

$$\boldsymbol{\Sigma} = \boldsymbol{R}\boldsymbol{S}\boldsymbol{S^T}\boldsymbol{R^T}. \tag{2}$$

The 3D covariance matrix $\boldsymbol{\Sigma}$ can be then projected onto 2D for the convenience of rendering each pixel:

$$\boldsymbol{\Sigma}' = \boldsymbol{J}\boldsymbol{W}\boldsymbol{\Sigma}\boldsymbol{W^T}\boldsymbol{J^T}, \tag{3}$$

where $\boldsymbol{\Sigma}'$ denotes the 2D covariance matrix, $\boldsymbol{J}$ is the Jacobian of the affine approximation of the projective transformation, and $\boldsymbol{W}$ is the viewing transformation matrix from the world to the camera coordinate frame. To render the color of a pixel on the image plane, we use the opacity $\sigma$ of the Gaussian and the spherical harmonics (SH) coefficients to generate 2D views using an $\alpha$-blending algorithm similar to the volumetric rendering in NeRF (Mildenhall et al., 2020). The rendering process is shown as follows:

$$C(p) = \sum_{i \in \mathcal{G}} T_i \alpha_i c_i, \tag{4}$$

$$T_i = \prod_{j=1}^{i-1}(1 - \alpha_j) \quad \text{and} \quad \alpha_i = \sigma_i e^{-\frac{1}{2}(p-\mu_i)^T \Sigma'(p-\mu_i)}, \tag{5}$$

where $C(p)$ denotes the color of a pixel located at $p$, $\mathcal{G}$ is a set of Gaussians along the camera ray sorted by depth with respect to the viewpoint, $T_i$ represents the transmittance, $c_i$ is the color of the Gaussian, and $\mu_i$ denotes the 2D coordinate of the 3D Gaussian projected onto the image plane. The detailed projection and rendering processes are described in Kerbl et al. (2023).

### 3.1.2 $N$-channel 3DGS

To develop a hyperspectral 3DGS framework that can synthesize novel views of hyperspectral images at $N$ different spectral bands, we extend the vanilla 3DGS framework (Kerbl et al., 2023) into an $N$-channel 3DGS ($N$-3DGS) to render images with $N$ wavelength bands. Two challenges are encountered when extending the 3DGS to render multi-channel hyperspectral images. Firstly, traditional structure-from-motion (SfM) methods such as COLMAP (Schonberger & Frahm, 2016) can only generate point clouds based on grayscale or RGB images. To solve this, we generate pseudo-RGB images from the hyperspectral images through the sensor simulation method in Chen et al. (2024a), and then feed them to COLMAP (Schonberger & Frahm, 2016) to obtain the sparse point clouds for initialization. Secondly, the vanilla 3DGS only supports rendering images in three channels, i.e., red, green, and blue. However, hyperspectral images contain more than three channels where each channel corresponds to a narrow, specific wavelength band across a continuous spectrum, often covering wavelengths from the visible to near-infrared regions. Based on the physical properties of the hyperspectral images, we modify the 3DGS algorithm by enabling the SH coefficients to render images with $N$ channels instead of 3 channels. Specifically, for each spectral band $i \in \{1, \ldots, N\}$, we associate a distinct set of spherical harmonics (SH) coefficients $\{SH_{i,lm}\}$ with each 3D Gaussian. The view-dependent radiance $c_i$ for channel $i$ and viewing direction $\mathbf{v}$ is then computed as:

$$c_i(\mathbf{v}) = \sum_{l=0}^{L}\sum_{m=-l}^{l} SH_{i,lm} \cdot Y_{lm}(\mathbf{v}), \tag{6}$$

where $Y_{lm}(\mathbf{v})$ denotes the real spherical harmonic basis function of degree $l$ and order $m$. This formulation allows each Gaussian to emit reflectance values in $N$ discrete wavelength bands, capturing fine-grained spectral reflectance variation as a function of viewing direction. These per-channel radiance values are then composited via alpha blending to generate hyperspectral renders. In this way, the designed 3DGS framework can be utilized to reconstruct 3D hyperspectral scenes and render novel view hyperspectral images.

### 3.1.3 Wavelength Encoder

While the 3DGS method can accurately capture the geometry of the hyperspectral scene, the reflectance values derived directly from the SH are suboptimal across different wavelengths. Notably, the fine-grained details are missing in the rendered images, especially at wavelengths beyond the visual range, This degradation arises from the lack of spectral priors in the vanilla 3DGS formulation—each wavelength channel is modeled independently, ignoring the smooth and correlated structure of natural reflectance spectra. As a result, the network struggles to produce consistent SH representations across the spectral dimension, leading to noisy or blurred outputs at certain wavelengths. To address this issue, we propose a wavelength encoder to learn wavelength-specific SH coefficients. Each wavelength in the hyperspectral images is firstly processed with a positional embedding module. This positional embedding module applies sinusoidal functions at multiple frequencies, enabling the learning of intricate wavelength-specific features. The positional embedding $\gamma$ for wavelength $\lambda$ is defined as:

$$\gamma(\lambda) = \left[\sin(2^0\pi\lambda), \cos(2^0\pi\lambda), \dots, \sin(2^{L-1}\pi\lambda), \cos(2^{L-1}\pi\lambda)\right]^\top. \tag{7}$$

where $L$ denotes the number of frequency bands used in the embedding, and thus determines the dimensionality of the resulting positional encoding.

To map these high-frequency features to the same dimension as the SH coefficients, we pass the positional embeddings through an MLP block, inspired by Yang et al. (Yang et al., 2024b). The MLP outputs wavelength-specific offsets, which are then added to the SH coefficients for each wavelength band. We use 3D spherical harmonics (SH), where each wavelength is represented by a set of SH coefficients encoding view-dependent color. In our case, using SH of degree 3 results in 16 coefficients per wavelength. The MLP is designed to match this dimensionality so that the offsets can be added directly to the base SH values. This allows the appearance of each Gaussian to adapt dynamically based on wavelength, improving spectral consistency and rendering quality.

$$\delta SH_\lambda = \mathrm{MLP}(\gamma(\lambda)), \tag{8}$$

$$SH_\lambda^+ = SH_\lambda + \delta SH_\lambda, \tag{9}$$

where $\delta SH_\lambda$ represents the offset of the SH value, $SH_\lambda$ and $SH_\lambda^+$ denote the SH of the 3D Gaussians at the given wavelength $\lambda$ before and after adding the offset. Notably, since other Gaussian parameters, such as position, rotation and scale, define the intrinsic geometry structure of the scene which are invariant across all the wavelengths, these geometric parameters are not modulated.

### 3.2 Hyperspectral Denoising with Diffusion Model

The wavelength encoder is able to capture wavelength-dependent SH coefficients for multi-channel hyperspectral images; however, the rendered images can still contain artifacts and noise. To further improve the quality of hyperspectral novel view synthesis, we integrate a diffusion-based denoising model directly into the 3DGS rendering pipeline and train the system in an end-to-end manner. Images rendered by 3DGS serve as the input for training the diffusion model, while the output from the diffusion model is used to compute the loss for training 3DGS. Unlike conventional approaches where diffusion operates as a post-processing step, our method allows the diffusion model to refine the raw hyperspectral renders during training, thereby jointly improving the training of the 3DGS itself. Given an initial hyperspectral render $\mathbf{X}_{\mathrm{3DGS}} \in \mathbb{R}^{H \times W \times N}$ from the 3DGS pipeline, we treat it as a noisy observation and feed it into the denoising diffusion model. The ground truth $\mathbf{X}_{\mathrm{GT}}$ is a raw captured noisy hyperspectral image. $\mathbf{X}_{\mathrm{GT}}$ is modeled as the output of a

reverse diffusion process conditioned on $\mathbf{X}_{\text{3DGS}}$. The forward process corrupts $\mathbf{X}_{\text{GT}}$ with Gaussian noise:

$$\mathbf{X}_t = \sqrt{\bar{\alpha}_t}\mathbf{X}_{\text{GT}} + \sqrt{1 - \bar{\alpha}_t}\boldsymbol{\epsilon}, \quad \boldsymbol{\epsilon} \sim \mathcal{N}(0, \mathbf{I}), \tag{10}$$

and the model learns to predict the noise $\boldsymbol{\epsilon}$ given the noisy input $\mathbf{X}_{\text{3DGS}}$ by minimizing the following loss:

$$\mathcal{L}_{\text{diffusion}} = \mathbb{E}_{\mathbf{X}_{\text{GT}}, t, \boldsymbol{\epsilon}} \left[ \|\boldsymbol{\epsilon} - \boldsymbol{\epsilon}_\theta(\mathbf{X}_t, t \mid \mathbf{X}_{\text{3DGS}})\|^2 \right], \tag{11}$$

where $\boldsymbol{\epsilon}_\theta$ is the predicted noise from the denoising network and $\mathbf{X}_t$ is the noisy input at timestep $t$. During training, the diffusion model implicitly learns to correct structured artifacts from the 3DGS renderings. This joint optimization allows the final rendered views to achieve high spectral fidelity while preserving the spatial realism of the 3D scene. The motivation behind using a diffusion model stems from the unique challenges of hyperspectral image reconstruction. Renders from 3DGS suffer from complex, structured artifacts, such as band-specific noise and spectral inconsistencies. Simple denoising models like autoencoder often struggle to correct without losing fine-grained details. However, diffusion models which iteratively refine data, are exceptionally well-suited for removing such structured noise while preserving the high-frequency spectral information. This observation motivates us to leverage diffusion model for the denoising of hyperspectral images. Please see Appendix. C.4 for quantitative comparison between diffusion model and autoencoder.

### 3.3 End-to-End Training

#### 3.3.1 Spectral Loss

Considering the physical property of hyperspectral images, where each pixel comprises a continuous emission spectra distribution, we argue that to achieve high-quality novel view synthesis, it is essential to ensure that the spectral distributions of the rendered views closely match those of the ground truth. To enforce this alignment, we introduce a spectral loss composed of two terms: a weighted Kullback-Leibler (KL) divergence and a cosine similarity penalty. These jointly encourage the model to produce hyperspectral outputs whose spectral distributions $D_\lambda$ accurately reflect those observed in the real scene.

Since KL divergence requires the input to be a probability distribution (i.e., non-negative and summing to 1), we normalize the predicted and ground-truth hyperspectral volumes using a softmax operation along the spectral axis. Specifically, for each pixel $(h, w), h \in H, w \in W$, we define the normalized spectral vector as:

$$\mathbf{D}_\lambda(h, w) = \text{softmax}(\mathbf{X}(h, w)), \tag{12}$$

where $\mathbf{X} \in \mathbb{R}^{H \times W \times N}$ is the unnormalized hyperspectral volume, and $\mathbf{X}(h, w) \in \mathbb{R}^N$ denotes the spectral vector at pixel $(h, w)$. The spectral loss is then computed as:

$$\mathcal{L}_{\text{spectral}} = \alpha \sum_{h=1}^{H} \sum_{w=1}^{W} \text{KL}\left(\mathbf{D}_\lambda^{\text{GT}}(h, w) \,\|\, \mathbf{D}_\lambda^{\text{pred}}(h, w)\right) + \beta \sum_{h=1}^{H} \sum_{w=1}^{W} \left(1 - \cos\left(\mathbf{D}_\lambda^{\text{GT}}(h, w), \mathbf{D}_\lambda^{\text{pred}}(h, w)\right)\right), \tag{13}$$

where $\mathbf{D}_\lambda^{\text{GT}}(h, w)$ and $\mathbf{D}_\lambda^{\text{pred}}(h, w)$ are the normalized spectral vectors from the ground truth and prediction at pixel $(h, w)$, respectively. Here, $\text{KL}(\cdot \| \cdot)$ denotes the Kullback–Leibler divergence, which penalizes distributional mismatches between the predicted and ground truth spectra, while $\cos(\cdot, \cdot)$ denotes the cosine similarity, which promotes angular alignment between the spectral vectors. The weights $\alpha$ and $\beta$ are user-defined hyperparameters controlling the relative importance of each term; in our experiments.

This spectral-aware formulation encourages the network to not only match the absolute intensities but also the shape and directionality of the spectral profiles, which is crucial for downstream applications like material classification or reflectance estimation.

#### 3.3.2 Overall Loss Function

To enable end-to-end training, we combine the vanilla 3DGS loss terms with the designed spectral and diffusion losses to form the final loss function $\mathcal{L}_{\text{HS-GS}}$, as shown below:

$$\mathcal{L}_{\text{HS-GS}} = w_1 \mathcal{L}_{\text{L1}} + w_2 \mathcal{L}_{\text{SSIM}} + w_3 \mathcal{L}_{\text{spectral}} + w_4 \mathcal{L}_{\text{diffusion}}. \tag{14}$$

In this equation, $\mathcal{L}_{\text{L1}}$ measures the pixel-wise difference between the rendered and ground truth images, and $\mathcal{L}_{\text{SSIM}}$ evaluates their structural similarity using the Structural Similarity Index Measure (SSIM). The spectral loss $\mathcal{L}_{\text{spectral}}$ enforces pixel-wise consistency between the predicted and ground truth spectral distributions. Finally, $\mathcal{L}_{\text{diffusion}}$ supervises the output of the diffusion model, which takes the noisy 3DGS render as input and refines it toward the clean hyperspectral target, correcting geometric distortions and spectral noise. The weights $w_1$, $w_2$, $w_3$, and $w_4$ are hyperparameters controlling the contribution of each term.

## 4  Experiments

### 4.1  Datasets

We evaluate our HS-GS framework on two hyperspectral datasets for agricultural scenes collected in Hyper-NeRF (Chen et al., 2024a), which are captured using two distinct hyperspectral imaging systems. BaySpec dataset is captured by a GoldenEye camera while Surface Optics dataset is captured by a SOC710-VP camera. Following the Hyper-NeRF benchmark (Chen et al., 2024a), we split each scene in both datasets into 90% and 10% images for training and testing, respectively. All evaluations are performed on the test sets. **BaySpec Dataset.** This dataset consists of hyperspectral images captured using a BaySpec GoldenEye camera, which provides a spatial resolution of $640 \times 512$ and a spectral resolution covering the range from 400 nm to 1100 nm across 141 narrow bands. The scenes feature three plant species: Anacampseros, Caladium and Pinecone. Each plant is placed on a motorized turntable and imaged from approximately 20 cm away. A total of 433 hyperspectral images were captured for each scene from a diverse set of viewpoints. **Surface Optics Dataset.** This dataset includes hyperspectral images captured using a Surface Optics SOC710-VP camera, which provides 128 spectral channels ranging from 370 nm to 1100 nm, with a spatial resolution of $696 \times 520$ pixels. Due to the camera's narrow field of view and shallow field of depth, it is mounted on a fixed tripod positioned about 2 m from the target. Two plant objects, Rosemary and Basil, are placed on a rotating table inside a Macbeth SpectraLight lightbooth to ensure consistent illumination.

### 4.2  Quantitative Results

We evaluate HS-GS on in total five plant scenes from the aforementioned two datasets. The implementation details and evaluation metrics are provided in the supplementary material. We compare our method against a carefully selected group of baselines spanning NeRF-based and 3D Gaussian Splatting (3DGS)-based methods, each chosen for their relevance to hyperspectral novel view synthesis. NeRF (Mildenhall et al., 2020) and 3DGS (Kerbl et al., 2023) serve as foundational models for radiance field rendering and explicit 3D representation, respectively. HyperNeRF (Chen et al., 2024a) is selected for its ability to model hyperspectral reflectances with spectral priors. MipNeRF (Barron et al., 2021) and MipNeRF360 (Barron et al., 2022) are included due to their strong performance in novel view synthesis tasks. Both leverage mipmapping and hierarchical sampling to effectively handle aliasing and unbounded scene geometry, making them robust candidates for high-fidelity view generation. TensoRF (Chen et al., 2022) is selected for its efficiency and compactness via tensor decomposition, which benefits rendering with high-dimensional outputs. Hyper-GS (Thirgood et al., 2025), a recent extension of 3DGS to hyperspectral rendering, represents the most competitive prior tailored specifically for this task. To ensure a fair comparison, all NeRF- and 3DGS-based baselines are extended to support $N$-channel hyperspectral outputs, allowing direct evaluation of both spatial reconstruction quality and spectral fidelity.

**BaySpec Results**  We present quantitative results of our method on the Pinecone, Caladium, and Anacampseros scenes from the BaySpec dataset in Table 1. HS-GS achieves the best overall performance on the Caladium and Anacampseros scenes, surpassing the cutting-edge methods such as Hyper-GS (Thirgood et al., 2025) and MipNeRF360 (Barron et al., 2022) On Pinecone, HS-GS achieves superior performance in SSIM and RMSE while remaining suboptimal in PSNR and SAM. In terms of rendering speed, our method is slower than vanilla 3DGS (Kerbl et al., 2023) due to the overhead introduced by the Wavelength Encoder and the Diffusion module, but achieves higher efficiency than the state-of-the-art hyperspectral 3DGS method (Thirgood et al., 2025). See Appendix C.3 for more details on memory usage, training time, and

Table 1: **Quantitative results on the BaySpec dataset.** We evaluate and compare HS-GS with both NeRF-based and 3DGS-based baselines on three hyperspectral scenes: Pinecone, Caladium, and Anacampseros, captured using a BaySpec GoldenEye camera. Our method achieves the state-of-the-art performance on most of the metrics across all the scenes. The best results are shown in **bold**, and the second best are underlined.

| Method | Pinecone | | | | Caladium | | | | Anacampseros | | | | FPS↑ |
|---|---|---|---|---|---|---|---|---|---|---|---|---|---|
| | PSNR↑ | SSIM↑ | SAM↓ | RMSE↓ | PSNR↑ | SSIM↑ | SAM↓ | RMSE↓ | PSNR↑ | SSIM↑ | SAM↓ | RMSE↓ | |
| NeRF (Mildenhall et al., 2020) | 22.82 | 0.6113 | 0.0446 | 0.0728 | 23.12 | 0.58348 | 0.0491 | 0.0709 | 24.12 | 0.6220 | 0.0384 | 0.0623 | 0.13 |
| MipNeRF (Barron et al., 2021) | 21.45 | 0.5738 | 0.0410 | 0.0856 | 23.36 | 0.5935 | 0.0487 | 0.0685 | 23.43 | 0.6160 | 0.0408 | 0.0786 | 0.09 |
| TensoRF (Chen et al., 2022) | 24.12 | 0.6454 | 0.0593 | 0.0625 | 24.79 | 0.6424 | 0.0516 | 0.0577 | 25.07 | 0.6569 | 0.0394 | 0.0558 | 0.17 |
| Nerfacto (Tancik et al., 2023) | 15.36 | 0.4935 | 0.0707 | 0.1709 | 20.67 | 0.6208 | 0.0529 | 0.0945 | 21.32 | 0.6423 | 0.0417 | 0.0867 | 0.50 |
| MipNeRF360 (Barron et al., 2022) | 25.93 | 0.7335 | **0.0279** | 0.0507 | 26.93 | 0.7371 | 0.0332 | 0.0461 | 26.73 | 0.7601 | 0.0230 | 0.0461 | 0.01 |
| Hyper-NeRF (Chen et al., 2024a) | 20.07 | 0.581 | 0.0725 | 0.1521 | 19.08 | 0.705 | 0.0533 | 0.0902 | 20.32 | 0.7260 | 0.0345 | 0.0789 | 0.47 |
| 3DGS (Kerbl et al., 2023) | 22.65 | 0.6039 | 0.0668 | 0.0819 | 23.50 | 0.7131 | 0.0289 | 0.0758 | 22.59 | 0.5786 | 0.0447 | 0.0853 | **78.10** |
| Hyper-GS (Thirgood et al., 2025) | **27.00** | 0.7509 | 0.0309 | 0.0447 | 27.70 | 0.8354 | 0.0271 | **0.0414** | 26.62 | 0.7545 | **0.0183** | 0.0460 | 2.31 |
| **HS-GS (Ours)** | 25.11 | **0.9347** | 0.0572 | **0.0244** | **27.86** | **0.9362** | **0.0224** | 0.0417 | **28.57** | **0.9490** | 0.0247 | **0.0381** | 2.43 |

Table 2: **Quantitative results on the Surface Optics dataset.** We evaluate and compare HS-GS with both 3DGS-based and NeRF-based baselines on two hyperspectral scenes: Rosemary and Basil, captured using a Surface Optics SOC710-VP camera. Our method surpasses Hyper-GS (Thirgood et al., 2025) on most of the metrics, and achieves the state-of-the-art on PSNR and RMSE on both scenes. Best results are **bolded**, and the second best are underlined.

| Method | Rosemary | | | | Basil | | | | FPS↑ |
|---|---|---|---|---|---|---|---|---|---|
| | PSNR↑ | SSIM↑ | SAM↓ | RMSE↓ | PSNR↑ | SSIM↑ | SAM↓ | RMSE↓ | |
| NeRF (Mildenhall et al., 2020) | 8.42 | 0.7461 | 0.0284 | 0.3560 | 9.91 | 0.5534 | 0.0796 | 0.5256 | 0.13 |
| MipNeRF (Barron et al., 2021) | 13.64 | 0.5684 | 1.0000 | 0.2083 | 11.01 | 0.5878 | 0.0728 | 0.5334 | 0.09 |
| TensoRF (Chen et al., 2022) | 12.10 | 0.7335 | 0.0212 | 0.2662 | 15.23 | 0.5811 | 0.0435 | 0.3628 | 0.20 |
| Nerfacto (Tancik et al., 2023) | 18.66 | 0.8366 | 0.0708 | 0.1025 | 16.54 | 0.7915 | 0.0176 | 0.1655 | 0.57 |
| MipNeRF360 (Barron et al., 2022) | 8.47 | 0.7518 | 0.0876 | 0.3825 | 13.92 | 0.8584 | 0.0497 | 0.2035 | 0.01 |
| Hyper-NeRF (Chen et al., 2024a) | 18.60 | 0.8870 | 0.0077 | 0.1187 | 16.81 | 0.7710 | 0.0170 | 0.1587 | 0.49 |
| 3DGS (Kerbl et al., 2023) | 25.56 | 0.9695 | 0.0028 | 0.0534 | 21.79 | 0.9385 | 0.0101 | 0.0897 | **79.00** |
| Hyper-GS (Thirgood et al., 2025) | 26.77 | **0.9845** | **0.0021** | 0.0445 | 25.30 | **0.9503** | 0.0051 | 0.0569 | 3.56 |
| **HS-GS (Ours)** | **28.54** | 0.9191 | 0.0043 | **0.0040** | **48.13** | 0.9340 | **0.0019** | **0.0018** | 2.95 |

inference efficiency evaluation. Overall, the experimental results demonstrate that HS-GS achieves the new state-of-the-art performance with high efficiency and generalizes well across diverse hyperspectral scenes.

**Surface Optics Results** We present quantitative results of our method on the Rosemary and Basil scenes from the Surface Optics dataset in Table 2. On Rosemary, HS-GS achieves leading performance with a PSNR of 28.54 and RMSE of 0.0040. On Basil, HS-GS obtains the best performance in PSNR, SAM and RMSE, indicating its precise spectral reconstruction and spatial detail preservation. For PSNR, our method surpasses the state-of-the-art method Hyper-GS (Thirgood et al., 2025) with around 22.83. We attribute this significant improvement to the following findings. 3DGS (Kerbl et al., 2023) and Hyper-GS (Thirgood et al., 2025) tend to introduce geometric artifacts in certain regions where severe structural variations and occlusions make spatial reconstruction less reliable. In contrast, our hyperspectral diffusion module addresses this issue by learning to correct these residual errors. Our model can enhance geometric fidelity and spectral consistency, enabling high-quality novel view synthesis even in scenes where 3DGS (Kerbl et al., 2023) and Hyper-GS (Thirgood et al., 2025) fall short. The quantitative results highlight the robustness of HS-GS in reconstructing 3D scenes with complex geometry and fine-grained spectral details.

### 4.3 Qualitative Results

In Figure 3, we present rendered hyperspectral images and difference heatmaps with the ground truth of our method and the baselines for three hyperspectral scenes: Caladium, Basil and Rosemary. We qualitatively compare our method with NeRF (Mildenhall et al., 2020), Hyper-NeRF (Chen et al., 2024a), and

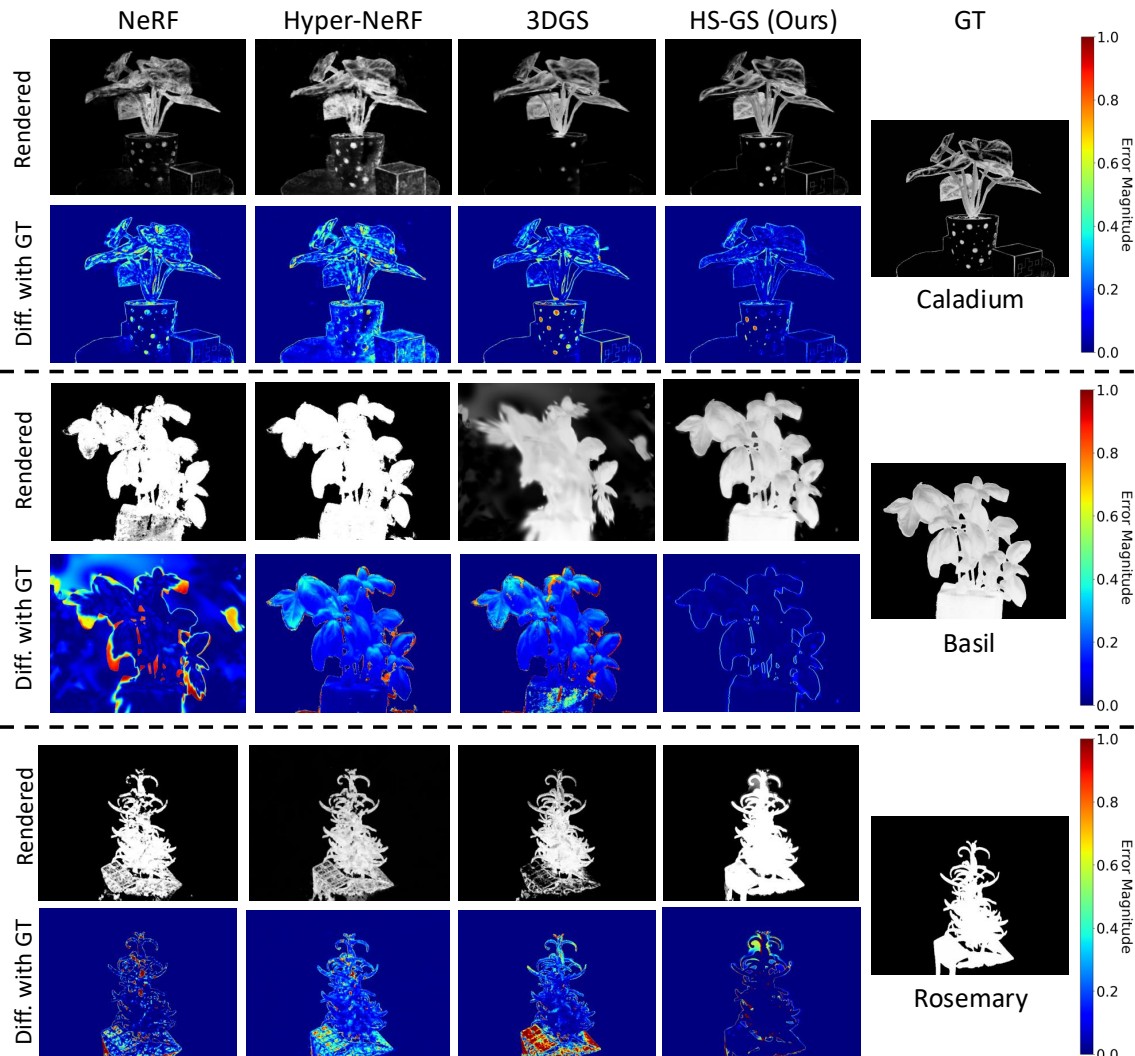

Figure 3: Qualitative comparison of novel view synthesis on the BaySpec and Surface Optics datasets. Caladium, Basil and Rosemary are visualized at the wavelength range of 750 nm to 768 nm. We demonstrate the rendered images and the difference heatmaps with the ground truth. We compare our method with NeRF (Mildenhall et al., 2020), Hyper-NeRF (Chen et al., 2024a) and 3DGS (Kerbl et al., 2023). HS-GS achieves superior performance in spatial reconstruction and spectral fidelity, with reduced artifacts and improved preservation of fine-grained details across both camera systems.

3DGS (Kerbl et al., 2023). We do not qualitatively compare ours with the most relevant work, Hyper-GS (Thirgood et al., 2025), because their code is not open-source. NeRF and Hyper-NeRF fail to recover fine-grained spatial details and suffer from high spectral reconstruction errors. 3DGS achieves finer structure recovering ability but still exhibits spectral inconsistencies. In contrast, our HS-GS significantly reduces reconstruction artifacts and reaches higher geometric accuracy across both hyperspectral camera systems.

To further examine our model's generalization ability to different spectral bands, we further provide Figure 4 to show the qualitative results on the Anacampseros scene across three spectral ranges: 400–418 nm (ultraviolet), 750–768 nm (near-infrared), and 1082–1100 nm (far-infrared). Notably, NeRF (Mildenhall et al., 2020) exhibits large deviations in its spectral curves due to overfitting to noise patterns inherent in hyperspectral data, leading to unstable and inaccurate spectral reconstructions. Hyper-NeRF (Chen et al., 2024a) alleviates this issue in the near-infrared range, but still suffers from severe spectral misalignment. 3DGS (Kerbl et al., 2023) demonstrates better performance, but fails to capture fine-grained spectral transitions. Compared to these baselines, our HS-GS achieves accurate spectral and spatial reconstruction across

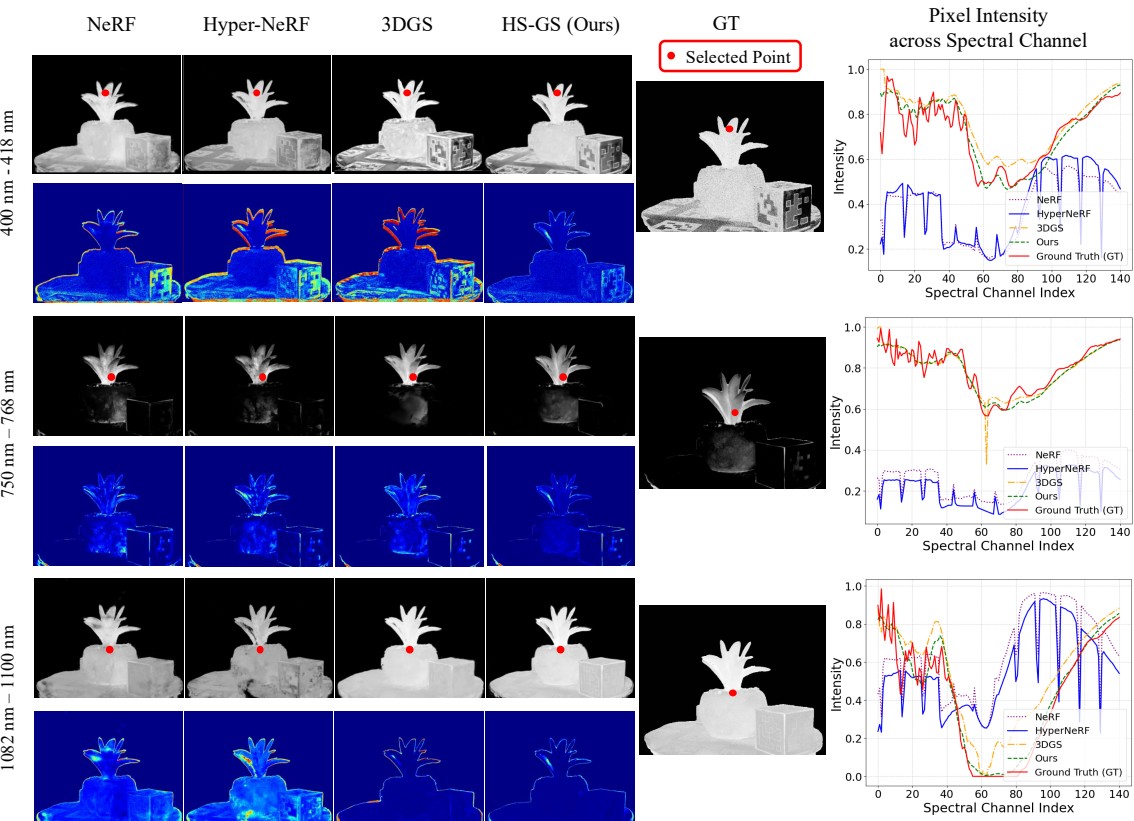

Figure 4: Qualitative results of the Anacampseros scene of three different wavelength ranges: 400 nm to 418 nm, 750 nm to 768 nm, 1082 nm to 1100 nm. The rendered images and difference heatmaps with the ground truth demonstrate the spectral fidelity and spatial consistency of the reconstruction results, particularly under challenging near-infrared and ultraviolet conditions. In addition, we demonstrate the reconstruction pixel intensity across all the spectral channels of three randomly selected points at the rightmost column. Compared to baselines, our method shows the highest degree of similarity with the ground truth.

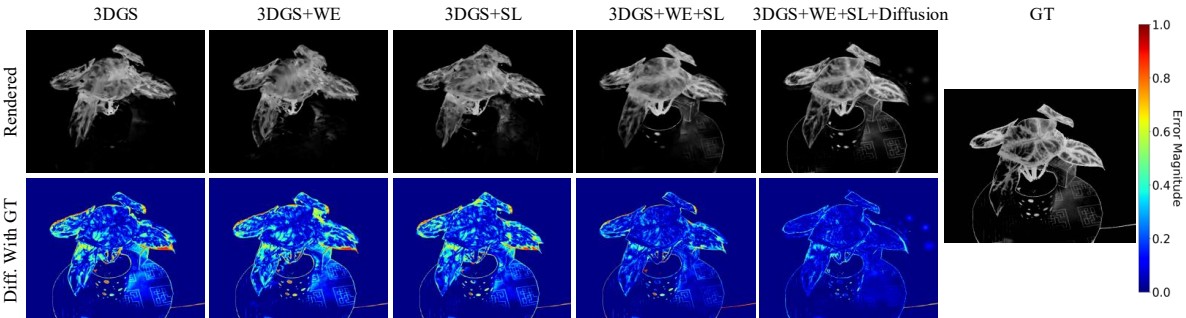

Figure 5: Qualitative ablation study on the Caladium scene. We show the rendered images and difference heatmaps with the ground truth. The proposed wavelength encoder and spectral loss are abbreviated as WE and SL respectively. With the addition of each designed module, the model gradually gets rid of detail artifacts and spectral distortions, achiving higher spatial and spectral reconstruction accuracy.

all the wavelength ranges, consistently aligning with the ground truth. We also plot the pixel intensity curves across spectral channel, as shown in the rightmost column of Figure 4. The pixel intensity of our method in green dashed line is closely matching the ground truth one in red solid line, and outperforms existing methods. We have added two videos showing the rendering quality across wavelengths for all our baselines.

## 4.4 Ablation Study

**Ablation Study of Main Components.** To understand the contribution of each component in HS-GS, we conduct an ablation study of the proposed modules, as shown in Table 3. With the wavelength encoder, our model can better capture spectral structure variations. With the spectral loss, our model can also recover spectral details in some regions. When applying both the wavelength encoder and the spectral loss, our model can yield more fine-grained reconstruction details with better spectral alignment. Furthermore, introducing the diffusion module to the model can facilitate the refinement of the geometry structure and de-noise the spectral outputs, especially in some challenging regions with sharp spectral transitions. Our full model can achieve top-performing performance across all the hyperspectral scenes. In addition, we examine the effect of diffusion model only, and find that it underperforms than our full model due to the lack of wavelength encoding and spectral-aware supervision. We also provide qualitative results of the ablation study on the Caladium scene in Figure 5. We can observe that the model progressively eliminates spectral distortions and spatial errors as each component is introduced.

**Effect of Diffusion Steps.** We examine the effect of varying the number of diffusion steps on the final performance. Compared to vanilla 3DGS (Kerbl et al., 2023), the application of hyperspectral diffusion module significantly improves both spectral accuracy (lower SAM) and visual sharpness (higher PSNR and SSIM). In Table 4, we quantitatively evaluate the performance of our model under different diffusion steps. By varying the number of diffusion steps, we observe that the model with 1000 steps consistently yields the best results across all the hyperspectral scenes. Additional ablation study results can be found in Appendix.

Table 3: **Ablation study of main components.** We examine the effect of the wavelength encoder, spectral loss, and hyperspectral diffusion model. Wavelength encoder and spectral loss are abbreviated as WE and SL, respectively. The reconstruction performance steadily improves as the components are applied.

| Method | SL | WE | Diffusion | Pinecone | | Anacampseros | | Caladium | |
|---|---|---|---|---|---|---|---|---|---|
| | | | | PSNR↑ / SSIM↑ | SAM↓ / RMSE↓ | PSNR↑ / SSIM↑ | SAM↓ / RMSE↓ | PSNR↑ / SSIM↑ | SAM↓ / RMSE↓ |
| 3DGS | | | | 21.40 / 0.8487 | 0.0912 / 0.0640 | 22.61 / 0.7622 | 0.0451 / 0.0682 | 20.40 / 0.8729 | 0.0583 / 0.0615 |
| 3DGS + SL | | ✓ | | 21.90 / 0.8485 | 0.0793 / 0.0579 | 22.77 / 0.7653 | 0.0365 / 0.0614 | 20.41 / 0.8720 | 0.0504 / 0.0571 |
| 3DGS+WE | ✓ | | | 21.71 / 0.8491 | 0.0761 / 0.0562 | 22.48 / 0.7652 | 0.0342 / 0.0598 | 20.38 / 0.8694 | 0.0493 / 0.0567 |
| 3DGS+WE+SL | ✓ | ✓ | | 22.18 / 0.8497 | 0.0715 / 0.0533 | 23.05 / 0.7703 | 0.0310 / 0.0542 | 20.66 / 0.8738 | 0.0458 / 0.0515 |
| **3DGS + Diffusion** | | | ✓ | 24.50 / 0.9285 | 0.0621 / 0.0292 | 27.10 / 0.9401 | 0.0264 / 0.0417 | 26.92 / 0.9263 | 0.0249 / 0.0439 |
| HS-GS | ✓ | ✓ | ✓ | 25.11 / 0.9347 | 0.0572 / 0.0244 | 28.57 / 0.9490 | 0.0247 / 0.0381 | 27.86 / 0.9362 | 0.0224 / 0.0417 |

Table 4: **Ablation study of the number of diffusion steps.** With the increase of diffusion steps, the model performance on all the scenes is steadily improved. We select 1000 diffusion steps as our final choice.

| Steps | Pinecone | | Anacampseros | | Caladium | |
|---|---|---|---|---|---|---|
| | PSNR↑ / SSIM↑ | SAM↓ / RMSE↓ | PSNR↑ / SSIM↑ | SAM↓ / RMSE↓ | PSNR↑ / SSIM↑ | SAM↓ / RMSE↓ |
| 10 | 24.91 / 0.9305 | 0.0624 / 0.0272 | 27.63 / 0.9418 | 0.0263 / 0.0435 | 27.42 / 0.9321 | 0.0271 / 0.0450 |
| 500 | 25.03 / 0.9332 | 0.0593 / 0.0251 | 28.02 / 0.9450 | 0.0250 / 0.0416 | 27.61 / 0.9349 | 0.0255 / 0.0420 |
| 1000 | 25.11 / 0.9347 | 0.0572 / 0.0244 | 28.57 / 0.9490 | 0.0247 / 0.0381 | 27.86 / 0.9362 | 0.0248 / 0.0312 |

## 5 Conclusion

In this work, we present Diffusion-Denoised Hyperspectral Gaussian Splatting (HS-GS), a novel framework for hyperspectral novel view synthesis that leverages a wavelength encoder to integrate wavelength-specific information and a spectral loss to enforce alignment with ground truth spectral distributions. To further enhance spectral fidelity and spatial realism, we incorporate a diffusion-based denoising model directly into the 3DGS rendering pipeline, treating intermediate hyperspectral renders as noisy observations and refining them end-to-end through a conditional reverse diffusion process. This joint optimization enables the model to correct structured artifacts such as band-wise noise and geometric distortions, resulting in high-quality, noise-resilient hyperspectral renderings. Extensive experiments demonstrate that HS-GS outperforms existing state-of-the-art methods, offering a robust and accurate approach to 3D hyperspectral scene reconstruction.

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

## A  Implementation Details

We implement our framework based on the 3D Gaussian Splatting implementation from Nerfstudio (Tancik et al., 2023), and train our model for 60,000 steps. The wavelength encoder is trained jointly with 3D Gaussians using an Adam optimizer (Kingma & Ba, 2015) with $\epsilon = 10^{-15}$. For 3D Gaussian parameters, we set the learning rate as $1.6 \times 10^{-4}$, while for the wavelength encoder, we set the learning rate as $1.6 \times 10^{-5}$. For the diffusion model, we leverage a timestep conditioned U-Net (Ronneberger et al., 2015; Ho et al., 2020) which is a hierarchical encoder-decoder structure with residual connections and time-step embeddings. The hyperspectral input is progressively downsampled through convolution blocks with increasing channel dimensions, and subsequently upsampled to reconstruct the output. We empirical set the weights $w_1$, $w_2$, $w_3$, and $w_4$ as 0.4, 0.2, 0.1 and 0.3 respectively to balance different loss terms. For spectral loss, we set $\alpha = 0.5$ and $\beta = 0.5$. All experiments are conducted on one NVIDIA A40 GPU.

## B  Evaluation Metrics

For comparison, we utilize the following metrics to quantitatively evaluate the model performance. Peak Signal-to-Noise Ratio (PSNR) measures pixel-level reconstruction quality. Structural Similarity Index Measure (SSIM) evaluates perceived structural similarity. Spectral Angle Mapper (SAM) quantifies spectral distortion in radians. Root Mean Squared Error (RMSE) captures absolute error between predicted and ground truth. Finally, Frames Per Second (FPS) denotes the number of rendered frames per second by the method.

## C  Ablation Study of Module Design

### C.1  Ablation Study of Positional Encoding Settings

As shown in Tab. 5 and Fig. 6, the absence of positional embeddings (No PE) results in a significant loss of geometric structure and reflective details, particularly in the central regions and beyond-visual-range wavelengths of the Pinecone scene. The rendered details appear blurred and fail to represent fine textures accurately. To isolate the effect of positional encoding, all experiments in this study were conducted without the diffusion module. Introducing positional embeddings ($L = 5$) significantly improves the rendering of finer reflective details and geometric accuracy, as evident in the sharper edges and clearer representation of reflective regions in the scene. However, further increasing the number of positional embeddings from $L = 5$ to $L = 10$ provides only marginal improvements, with relatively small enhancements in continuity and fidelity. This highlights that while positional embeddings are critical for wavelength encoding, increasing them beyond a certain threshold yields diminishing returns in terms of rendering quality.

Table 5: **Ablation study on the impact of positional encoding in HS-GS.** We evaluate the effect of different positional encoding settings without the diffusion module to isolate the contribution of the wavelength encoder. Performance is reported on the Pinecone, Anacampserous, and Caladium scenes from the Hyperspectral-NeRF dataset (Chen et al., 2024a), along with average results.

| Method | Pinecone | | Anacampserous | | Caladium | | Average | |
|---|---|---|---|---|---|---|---|---|
| | PSNR↑ | SSIM↑ | PSNR↑ | SSIM↑ | PSNR↑ | SSIM↑ | PSNR↑ | SSIM↑ |
| No PE | 18.31 | 0.8424 | 22.84 | 0.7681 | 20.18 | 0.8691 | 20.44 | 0.8265 |
| $L = 5$ | 22.13 | 0.8496 | 23.04 | **0.7716** | 20.54 | 0.8732 | 21.90 | **0.8315** |
| $L = 10$ **(Ours)** | **22.18** | **0.8497** | **23.05** | 0.7703 | **20.66** | **0.8738** | **21.96** | 0.8313 |

### C.2  Ablation Study of Spectral Loss Weight

As shown in Tab. 6 and Fig. 7, the choice of the spectral loss (SL) weight has a significant impact on rendering quality. To isolate the effect of spectral loss, all experiments in this section are conducted without

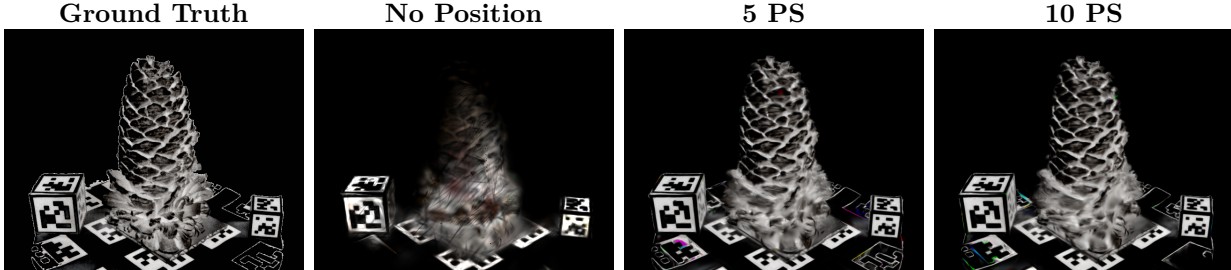

| Ground Truth | No Position | 5 PS | 10 PS |

Figure 6: **Comparison of positional encoding configurations in HS-GS without diffusion.** The ground truth (left) is shown alongside variants of the wavelength encoder without positional embedding, with 5-dimensional embeddings, and with 10-dimensional embeddings. All models are trained without the diffusion module to isolate the effects of wavelength encoding.

the diffusion model. When the SL weight is set to 0.1, the rendered details in the central portion of the Pinecone plant are visible but lack refinement, and the reflective properties are not accurately captured. Increasing the SL weight to 0.2 leads to a noticeable improvement in rendering accuracy. The finer details, particularly in the central portion, are better defined, and the reflective regions exhibit improved fidelity. This demonstrates that a moderate SL weight effectively balances spectral accuracy and perceptual quality. However, further increasing the SL weight to 0.3 diminishes returns and degrades rendering quality. The central portion of the Pinecone plant appears overly darkened, with some fine details becoming obscured. This suggests that an excessively high SL weight overemphasizes spectral accuracy at the cost of spatial and perceptual fidelity.

Table 6: **Ablation study of spectral loss weight $w_3$ in HS-GS.** We evaluate the impact of varying the spectral loss weight $w_3$ on model performance across three scenes from the Hyperspectral-NeRF dataset (Chen et al., 2024a). The results show that $w_3 = 0.2$ yields the best performance and is used in our final model.

| $w_3$ | Pinecone | | Anacampserous | | Caladium | | Average | |
|---|---|---|---|---|---|---|---|---|
| | PSNR↑ | SSIM↑ | PSNR↑ | SSIM↑ | PSNR↑ | SSIM↑ | PSNR↑ | SSIM↑ |
| 0.1 | 22.06 | 0.8493 | 22.99 | 0.7692 | 20.56 | 0.8731 | 21.87 | 0.8305 |
| 0.3 | 22.05 | 0.8497 | 22.81 | 0.7673 | 20.56 | 0.8744 | 21.81 | 0.8305 |
| **0.2 (Ours)** | **22.18** | **0.8497** | **23.05** | **0.7703** | **20.66** | 0.8738 | **21.96** | **0.8312** |

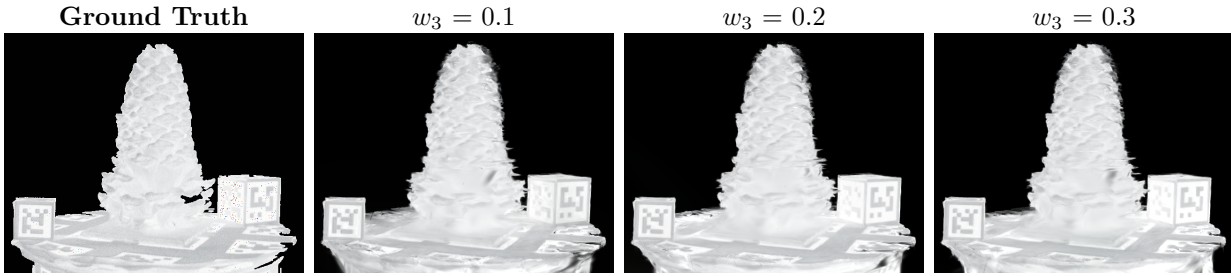

| Ground Truth | $w_3 = 0.1$ | $w_3 = 0.2$ | $w_3 = 0.3$ |

Figure 7: **Comparison of spectral loss (SL) results for different configurations on Pinecone.** The ground truth (left half) is shown alongside the right halves of $w_3 = 0.1$, $w_3 = 0.2$, and $w_3 = 0.3$ configurations.

## C.3 Runtime and Memory Benchmarks

We provide a detailed evaluation and comparison of the runtime and memory consumption of HS-GS and other methods. Table 7 reports training time per iteration, total memory usage, number of Gaussians

and inference speed in frames per second. We observed that the vanilla 3DGS model is the most efficient, achieving inference speeds above 75 FPS. The inclusion of the diffusion module increases memory usage and reduces inference speed to around 2.4 FPS. Despite this, HS-GS still performs much faster than NeRF-based methods in both training and rendering speed. In sum, HS-GS achieves a good balance between reconstruction performance and computational efficiency.

Table 7: **Runtime and memory benchmarks.** We report training iteration time (s), peak GPU memory (GB), number of Gaussians and FPS.

| Method | Train. Time (s) | GPU Mem (GB) | Gaussians | FPS (↑) |
|---|---|---|---|---|
| NeRF | 821 | 10.9 | - | 0.13 |
| Hyper-NeRF | 261 | 10.6 | - | 0.5 |
| 3DGS | 0.48 | 7.6 | 112,350 | 75.2 |
| 3DGS + Diffusion | 1.23 | 19.1 | 134,024 | 2.4 |
| 3DGS + Spectral Loss | 0.85 | 11.8 | 128,910 | 54.3 |
| 3DGS + Wavelength Encoder | 0.87 | 12.2 | 129,704 | 52.7 |
| HS-GS (Ours) | 1.25 | 19.4 | 134,472 | 2.2 |

Table 8: **Performance comparison between HS-GS and autoencoder baseline on the BaySpec dataset.** HS-GS consistently outperforms 3DGS, 3DGS+Diffusion and autoencoder in spatial and spectral fidelity, as measured by PSNR, SSIM, RMSE, and SAM.

| Method | Scene | PSNR ↑ | SSIM ↑ | RMSE ↓ | SAM ↓ |
|---|---|---|---|---|---|
| Autoencoder | Pinecone | 10.45 | 0.2841 | 0.2763 | 0.1984 |
| | Anacampseros | 9.90 | 0.2291 | 0.3068 | 0.2145 |
| | Caladium | 11.02 | 0.3187 | 0.2634 | 0.1862 |
| 3DGS | Pinecone | 21.40 | 0.8487 | 0.0640 | 0.0912 |
| | Anacampseros | 22.61 | 0.7622 | 0.0682 | 0.0451 |
| | Caladium | 20.40 | 0.8729 | 0.0615 | 0.0583 |
| 3DGS + Diffusion | Pinecone | 24.50 | 0.9285 | 0.0292 | 0.0621 |
| | Anacampseros | 27.10 | 0.9401 | 0.0417 | 0.0264 |
| | Caladium | 26.92 | 0.9263 | 0.0439 | 0.0249 |
| HS-GS (Ours) | Pinecone | **25.11** | **0.9347** | **0.0244** | **0.0572** |
| | Anacampseros | **28.57** | **0.9490** | **0.0381** | **0.0247** |
| | Caladium | **27.86** | **0.9362** | **0.0417** | **0.0224** |

### C.4 Autoencoder Baselines on BaySpec Dataset

To evaluate the performance of traditional low-dimensional latent reconstruction models on hyperspectral data, we train an autoencoder and a variational autoencoder on three representative plant scenes (e.g.: Pinecone, Anacampseros, Caladium) from the BaySpec dataset. The results are presented in Table. 8.

**Autoencoder:** We firstly train an autoencoder baseline using a U-Net-style encoder-decoder architecture. The autoencoder baseline consistently underperforms in both spatial and spectral metrics (e.g.: PSNR, SSIM, SAM), demonstrating its inferior performance on preserving fine-grained spectral features across view-dependent geometry and material variations. Notably, SAM scores are significantly worse than those of 3DGS or our proposed HS-GS method, validating that simple bottleneck-based reconstructions are inadequate for hyperspectral consistency.

**Variational Autoencoder:** In addition, we train a variational autoencoder (VAE) baseline. However, due to the high channel dimensionality of hyperspectral inputs which are up to 141 bands, the VAE's latent

sampling and reconstruction pipeline ran out of memory in our experiment. This further reinforces the need for spatially conditioned and spectrally aware architectures, such as HS-GS, that can scale to such domains.

### C.5  Relation to Prior 3DGS + Diffusion Works

While our approach integrates 3D Gaussian Splatting with a diffusion model, it differs significantly from prior works such as **GaussianObject** (Yang et al., 2024a) and **MVSplat360** (Chen et al., 2024b) in both motivation and methodology.

**GaussianObject** focuses on object-level scene synthesis from sparse views and employs a frozen, pre-trained 2D diffusion model as a generative decoder to enhance rendering realism. In contrast, our diffusion model is trained in an end-to-end manner and designed specifically for denoising hyperspectral radiance outputs. Compared to GaussianObject, HS-GS directly learns to correct spectral distortions aligned with wavelength-dependent SH encoding and spectral loss, rather than enhancing RGB textures. GaussianObject operates purely in the natural image domain, without any consideration for hyperspectral fidelity.

**MVSplat360** addresses novel view synthesis for outdoor scenes, leveraging multi-view Gaussian splatting and a depth-aware neural renderer. While it incorporates a diffusion model to enhance RGB image quality, it is also pre-trained and only focusing on natural image restoration. HS-GS, in contrast, deals with hyperspectral radiance modeling and directly incorporates spectral priors through wavelength-specific SH modulations, KL-based spectral loss, and a custom diffusion denoiser tuned for spectral domains.

To the best of our knowledge, HS-GS is the first system to jointly train a wavelength-aware 3DGS model with a diffusion module tailored for high-fidelity hyperspectral scene reconstruction. It integrates spectral-domain learning objectives and view-aware spatial consistency to address the unique challenges of HSI.

