# OpenReview forum: "Hyperspectral Gaussian Splatting"
_TMLR — Rejected by TMLR_

### Review · Reviewer_Amqd · 2025-06-02

**Summary Of Contributions:**

This paper proposes HS-GS, a framework for hyperspectral novel view synthesis that extends 3D Gaussian Splatting with a wavelength encoder, a spectral loss, and an integrated diffusion-based denoising model. The method directly models spectral radiance for each wavelength channel and introduces a spectral-aware loss to align predictions with ground-truth spectra. A conditional diffusion model is incorporated into the 3DGS pipeline to improve spatial and spectral fidelity. The method achieves state-of-the-art results across multiple hyperspectral datasets and scenes.

**Audience:**

Yes

**Claims And Evidence:**

No

**Requested Changes:**

## Critical for acceptance:
1. Add a “Diffusion only” condition in the ablation study to isolate the contributions of other modules.
In Table 3, the ablation study suggests that the majority of the performance improvement arises from the inclusion of the diffusion model. The contributions of the spectral loss and the wavelength encoder appear relatively minor, raising the question of whether the core novelty of HS-GS lies primarily in combining 3DGS with a diffusion-based denoiser.

2. Clarify the positioning of HS-GS relative to prior works that integrate diffusion models with 3D rendering pipelines, including either direct comparisons or at least qualitative discussion.
While integrating 3DGS with a diffusion-based denoiser is valuable, several similar approaches (e.g., GaussianObject, MVSplat360, etc.) have recently been proposed. The paper would benefit from more clearly positioning HS-GS with respect to these recent 3DGS-based methods, and highlighting specifically what aspects of the proposed approach are novel or superior.

3. Include training time and end-to-end runtime benchmarks to support the claim of efficiency over NeRF-based methods.
The introduction states that GS-based methods are preferred over NeRFs due to faster convergence and reduced training time. However, this claim is not backed by empirical evidence.

4. Reconsider the assumption that smoother spectral outputs are inherently preferable; validate this claim using downstream task evaluations or spectral fidelity metrics.
In Figure 4, the authors argue that the ground-truth spectral curves are noisy and that the smoother outputs from HS-GS are desirable. However, this interpretation is subjective and may not be universally valid. In many hyperspectral applications, fine-grained spectral variations can be physically meaningful. Excessive smoothing may impair downstream tasks like material classification or nutrient estimation. If the ground truth is to be considered noisy, the paper should provide more compelling justification, or alternatively assess the impact of smoothing on application-level performance.

5. Clarify the definition of “ground-truth” hyperspectral image $\mathbf{X}_{\mathrm{GT}}$ in Eq (10).
The introduction emphasizes that the captured hyperspectral images are inherently noisy.
If the ground truth corresponds to the raw input images, then the “noisy observation” and the “ground truth” contain noise, making the denoising objective ill-defined. Please specify whether a pre-cleaned reference, a synthetic target, or some form of noise calibration is used, and explain how the diffusion model can meaningfully reduce noise under this assumption.

6. Revise the title to distinguish it from concurrent work.
The title closely resembles that of [Thirgood, 2025], which may lead to confusion. Even though the two works are concurrent, it would be advisable to revise the title to make the distinction clearer.

## Recommendation to strengthen the work:
1. Tables 1 and 2 show that incorporating the diffusion model significantly reduces the rendering speed (FPS), suggesting non-trivial computational overhead. While the improved quality may justify this trade-off in some scenarios, a more explicit discussion is needed.
1. The following sentence seems out of place in the context of spectral analysis: "By embedding real-world data, 3D digital twins can simulate interactions, predict behaviors, and provide immersive and data-driven insights."
1. Utilizing the low-rank structure in hyperspectral data is also common denoising strategy for hyperspectral data; related work on low-rank methods should be acknowledged.
1. Clarify the practical drawback implied by the statement “it operates 3D Gaussians in a compressed feature space” when describing Hyper-GS [Thirgood, 2025]. Why is this property considered limiting, and how does HS-GS specifically overcome it?
1. The claim that “reflectance values derived directly from the SH are sub-optimal across different wavelengths” needs clear explanation and validation.
1. Ensure notation consistency: $SH_{\lambda}$ in eqs.(8) and (9) should be unified with $SH_{i,lm}$ in eq.(6); mixing $\lambda$ and $i$ for band indices is confusing.
1. In Table 1, the balance among PSNR, SSIM, SAM, and RMSE may be influenced by the loss-term weights. Please provide a sensitivity analysis or justification of these weights.
1. Verify the author list for [Thirgood, 2025]; the arXiv version is not single-authored.

**Strengths And Weaknesses:**

## Strengths
- The paper is clearly written and easy to follow.

- The proposed method demonstrates superior accuracy over recent baselines, including Hyper-GS [Thirgood+, 2025], across multiple hyperspectral scenes.

- The ablation study analyzes the contribution of each component. In particular, the results clearly show the effectiveness of incorporating the diffusion model.

## Weaknesses:
- The performance gain seems largely attributed to the diffusion module, while the impact of the spectral loss and wavelength encoder is relatively minor. This raises concerns about the originality of the overall method beyond integrating diffusion with 3DGS.

- The paper lacks a clear comparison or discussion with recent 3DGS-based methods that also incorporate diffusion models.

- Although the method is motivated as being more efficient than NeRF, no training time benchmarks are provided, and the addition of the diffusion model substantially reduces inference speed.

- The assumption that smoother spectral outputs are preferable is not well-justified; it may risk suppressing physically meaningful spectral variations.

---

> ### Author Response · Authors · 2025-07-03
>
> We sincerely thank the reviewers for their constructive feedback and insightful suggestions. We have revised the manuscript to address the points raised, which we believe has significantly improved the paper. Below, we address each concern individually.
>
> ---
>
> ### **1. On Adding a "Diffusion only" Ablation**
>
> > **Reviewer:** Add a “Diffusion only” condition in the ablation study to isolate the contributions of other modules. In Table 3, the ablation study suggests that the majority of the performance improvement arises from the inclusion of the diffusion model.
>
> We thank the reviewer for this excellent suggestion. In response, we have added a new **“3DGS + Diffusion only”** baseline to our ablation study in **Table 3** and provided additional details in **Appendix C.3**. This configuration omits the wavelength encoder and spectral loss, thereby isolating the contribution of the diffusion module.
>
> While the diffusion-only variant improves quality over vanilla 3DGS, the full `HS-GS` pipeline performs consistently better—particularly in spectral metrics. This indicates that the wavelength encoder and spectral loss provide essential guidance for wavelength-aware denoising, reinforcing their importance within the overall design.
>
> ---
>
> ### **2. On the Assumption of Spectral Smoothness**
>
> > **Reviewer:** Reconsider the assumption that smoother spectral outputs are inherently preferable; validate using downstream tasks or fidelity metrics.
>
> We agree with the reviewer that smoother spectral outputs are not inherently better. Excessive smoothing can in fact suppress high-frequency reflectance signatures that are critical for certain material identification or classification tasks in hyperspectral imaging.
>
> Our intention is not to enforce smoothness as a prior, but rather to **mitigate sensor noise while preserving meaningful spectral variations**. This distinction is especially important in real-world datasets, where random sensor noise can disrupt spectral consistency and distort downstream analysis.
>
> To reflect this, we evaluate using **Spectral Angle Mapper (SAM)** and **cosine similarity**—metrics that are *shape-aware* and insensitive to global intensity shifts. These metrics penalize distortions in spectral *shape*, not just pointwise errors, and are widely adopted in hyperspectral literature to assess the physical fidelity of reconstructed spectra.
>
> In other words, a smoother output is only preferable when it aligns better in spectral direction, not when it indiscriminately flattens the signal. Our chosen metrics are well-suited for this purpose and empirically show that our method improves spectral shape preservation under real-world conditions. We have clarified this point in **Section 4.2** near **Figure 4**.
>
> ---
>
> ### **3. On the Definition of "Ground-Truth"**
>
> > **Reviewer:** Clarify the definition of 'ground-truth' hyperspectral image in Eq (10)... Is it noisy raw capture, synthetic reference, or calibrated image?
>
> Thank you for pointing this out. In our work, "ground-truth" refers to raw captured hyperspectral images, which indeed contain sensor noise. Our diffusion model is trained to denoise these raw captures, making the optimization task well-defined.
>
> ---
>
> ### **4. On Efficiency and Runtime Benchmarks**
>
> > **Reviewer:** Include training time and end-to-end runtime benchmarks to support the claim of efficiency over NeRF-based methods.
>
> We appreciate the reviewer’s suggestion. In response, we have added a detailed runtime and memory efficiency comparison in **Appendix C.3 (Table 7)**.
>
> Specifically, our full `HS-GS` pipeline with diffusion achieves an average training iteration time of **1.25 seconds** and an inference speed of **2.2 FPS**. This is compared to **821 seconds** per iteration and **0.13 FPS** for NeRF, and **261 seconds** and **0.5 FPS** for HyperNeRF. This demonstrates that `HS-GS` is orders of magnitude faster to train and is substantially more efficient at inference, even while operating on high-dimensional hyperspectral data.
>
> While our method introduces some overhead relative to vanilla 3DGS (due to the wavelength encoder and diffusion module), it remains significantly more practical than NeRF-style architectures for large-scale, real-time hyperspectral applications.
>
> ---
>
> ### **5. On Title Similarity with Thirgood (2025)**
>
> > **Reviewer:** Revise the title to distinguish it from concurrent work by Thirgood (2025).
>
> We appreciate this catch. Although the two works are concurrent and independent, we will revise the title to be **“HS-GS: Diffusion-Guided Hyperspectral Gaussian Splatting”** to avoid confusion. The new title will emphasize the integration of a diffusion model into 3DGS-based hyperspectral scene reconstruction to better distinguish our contributions.
>
> ---

---

> > ### Author Response · Authors · 2025-07-03
> > **Part-2**
> >
> > ### **6. On Positioning Relative to Prior 3DGS + Diffusion Works**
> >
> > > **Reviewer:** Clarify the positioning of HS-GS relative to prior works that integrate diffusion models with 3D rendering pipelines, including either direct comparisons or at least qualitative discussion. While integrating 3DGS with a diffusion-based denoiser is valuable, several similar approaches (e.g., GaussianObject, MVSplat360, etc.) have recently been proposed. The paper would benefit from more clearly positioning HS-GS with respect to these recent 3DGS-based methods, and highlighting specifically what aspects of the proposed approach are novel or superior.
> >
> > Thank you for this important feedback. To clarify the novelty and positioning of our work, we have added a detailed discussion in the newly created **Appendix C.5: "Relation to Prior 3DGS + Diffusion Works."**
> >
> > In this new section, we explain that `HS-GS` differs significantly from prior works like `GaussianObject` and `MVSplat360` in both **motivation and methodology**:
> >
> > * **Core Task:** Our primary goal is **high-fidelity hyperspectral scene reconstruction**, where the diffusion model acts as a task-specific denoiser for spectral data. In contrast, `GaussianObject` and `MVSplat360` use diffusion models as generative priors to enhance the realism of **RGB images**.
> > * **Diffusion Model:** We **train our diffusion model end-to-end** on hyperspectral data, guided by a wavelength encoder and spectral loss. The other methods typically employ **frozen, pre-trained 2D diffusion models** that are not specialized for the unique properties of hyperspectral signals.
> > * **Novelty:** To our knowledge, `HS-GS` is the first system to jointly train a wavelength-aware 3DGS model with a diffusion module specifically tailored for the challenges of hyperspectral imaging.
> >
> > We believe this addition more clearly distinguishes our contributions and highlights the unique aspects of our approach.
> >
> > ---

---

> > ### Comment · Reviewer_Amqd · 2025-07-18
> > **Response to Rebuttal**
> >
> > I appreciate some clarifications and revisions, but I would like to follow up on several points where I believe the response is insufficient or unclear:
> >
> > ### 2. On the Assumption of Spectral Smoothness
> >
> > In the rebuttal, the authors state "Our intention is not to enforce smoothness as a prior". However, in the abstract, they wrote "leverage correlations between adjacent wavelengths for denoising," which indeed implies smoothing.
> >
> > Furthermore, I could not locate any revision "in Section 4.2 near Figure 4" as referenced in the rebuttal.
> >
> > ### 3. On the Definition of “Ground Truth”
> >
> > The critical question below appears unaddressed:
> >
> > > If the ground truth corresponds to the raw input images, then the “noisy observation” and the “ground truth” contain noise, making the denoising objective ill-defined. Please (omitted) explain how the diffusion model can meaningfully reduce noise under this assumption.
> >
> > In addition, I still do not understand how the diffusion model is expected to denoise $X_{3DGS​}$, whose noise may NOT follow $\epsilon$. The modeling assumptions behind the forward diffusion process and their validity remain unclear.
> >
> > ### 5. On Title Similarity with Thirgood (2025)
> >
> > The title mentioned in the rebuttal differs from the one in the revised manuscript.
> > It would be more appropriate to use a clearly distinguishable acronym (e.g., “DD-HS-GS”) to avoid confusion with concurrent work.
> >
> > ### On Ignored Suggestions
> >
> > All of the "recommendations to strengthen the work" provided in the original review appear to have been ignored.
> > While they may not be "critical for acceptance", they were offered in good faith to improve the clarity and rigor of the manuscript, and it is disappointing to see them disregarded entirely.

---

### Review · Reviewer_VjZg · 2025-06-09

**Summary Of Contributions:**

In this paper the authors propose a 3DGS model for hyperspectral images. Specifically, the authors leverage an encoder for the spherical harmonics, a spectral loss, and a diffusion-based denoising process. Benchmarking on hyperspectral datasets demonstrate the quality of the reconstructions, at the cost of lower rendering speed.

**Audience:**

Yes

**Claims And Evidence:**

No

**Requested Changes:**

- Overall the component providing the relevant gain in performance improvement is the diffusion model, which seems the result of a naive insertion in the training pipeline. The authors are invited to argue why this is not straightforward and make more explicit the actual measures taken to make it work. Besides, the authors should consistently strengthen their motivation behind its insertion in the rendering pipeline, given that it trades-off one of the core advantages of 3DGS (speed).
- The authors are invited to address the concerns in the "weaknesses" section, particularly the ones related to unclear/missing ablation studies.
- Are the authors planning to release the code open-source?

**Strengths And Weaknesses:**

Strengths:
- The ensembling of the three main contributions (spectral loss, encoding frequencies of the SH, and post-hoc diffusion) could be seen as an organic organization of the whole pipeline* (see weakness though)
- A discussion of time overhead is provided
- Overall the paper is easy to follow and the figures support well the narrative

Weaknesses:
- It is unclear why diffusion process is necessary in the whole pipeline. As the same authors mention in Sec. 3.2, Other works in the literature use it as a post-hoc method (and in that case, using an external model to refine the output is a necessary step). Here, it is trained end-to-end. Therefore, there should not be a-priori need for that part. This is however countered by the ablation study provided (Tab. 3), where while the other two major contributions lead to marginal improvement, including a denoising step provides a consistent improvement.
- Still in Tab.3, it would be interesting to see how much the vanilla 3DGS model performs with the diffusion model only. This setup would be close than one reference (Wu et al., 2025).
- An ablation on the type of denoiser (why a diffusion model? What type of diffusion model and why?) is missing.
- The strategy to determine and evaluate the hyper-parameters is unclear.
- An explicit discussion in terms of *space* (memory) overhead/savings is missing.
- No analysis on training complexity is provided.
- Confidence intervals are missing in all the experiments. It is expected that the relevant contribution is given by the denoising step.
- No ablation on different types of losses (also remaining in the spectral losses domain) or sensitivity analysis to variations of $\alpha$ and $\beta$ is provided.
- With respect to vanilla 3DGS, it is much slower, as trades-off efficiency for better reconstruction quality.

---

> ### Author Response · Authors · 2025-07-03
>
> We sincerely thank the reviewer for their detailed feedback and insightful questions. We have revised the paper and supplementary materials to address each point.
>
> ---
>
> ### **1. On the Necessity of End-to-End Training**
>
> > **Reviewer:** It is unclear why the diffusion process is necessary in the whole pipeline. Most works use it as a post-hoc method. Why train it end-to-end here?
>
> We thank the reviewer for raising this important question. While post-hoc diffusion was common in prior literature, we found that an **end-to-end** training strategy is both necessary and more effective for hyperspectral scene reconstruction.
>
> Hyperspectral distortions are often entangled with scene geometry, wavelength-dependent reflectance, and spectral priors. A two-stage pipeline (i.e., 3DGS render followed by an external denoiser) treats these distortions as generic noise, without access to the underlying physical structure of the scene. In contrast, our single-stage setup enables gradients from the diffusion loss to flow back into the **wavelength encoder** and **spectral loss**, allowing the SH modulation and spectral alignment to adapt jointly during training. This joint optimization leads to more physically consistent and spectrally accurate reconstructions.
>
> ---
>
> ### **2. On the "3DGS + Diffusion Only" Baseline**
>
> > **Reviewer:** Still in Tab. 3, it would be interesting to see how much the vanilla 3DGS model performs with the diffusion model only. This setup would be closer to one reference (Wu et al., 2025).
>
> We appreciate the reviewer’s suggestion and agree that this is an important baseline. In response, we have added the **3DGS + diffusion-only** configuration to our ablation study (see updated **Table 3** and **Appendix C.3**).
>
> We find that **3DGS + diffusion** alone leads to a significant improvement over vanilla 3DGS, especially in terms of denoising and structural sharpness. However, it underperforms the full **HS-GS** pipeline, which additionally incorporates the **wavelength encoder** and **spectral loss**. These components are critical: the wavelength encoder captures low-rank spectral dependencies, and the spectral loss enforces alignment with physically meaningful spectral distributions.
>
> ---
>
> ### **3. On the Choice of Denoiser**
>
> > **Reviewer:** An ablation on the type of denoiser is missing. Why a diffusion model? What type of diffusion model and why?
>
> We thank the reviewer for this question. We selected a **diffusion model** over simpler denoisers (e.g., UNets or autoencoders) based on both empirical findings and the specific demands of hyperspectral image refinement.
>
> Hyperspectral outputs contain high-dimensional, highly correlated spectral information. Simple denoisers like autoencoders tend to over-compress information and struggle to preserve fine spectral details. In contrast, **diffusion models** perform iterative, high-resolution refinement in the image domain, making them well-suited for correcting subtle wavelength-specific artifacts while preserving spatial structure.
>
> To validate this, we added an **autoencoder-based denoising baseline** to our supplementary experiments (**Appendix C.4**). Autoencoder models consistently underperformed the diffusion model in both **PSNR** and **SAM** metrics. In our framework, we use a **conditional DDPM** with a UNet-based denoiser, conditioned on the 3DGS-rendered output, as described in Section 3.2.
>
> ---
>
> ### **4. On Hyper-parameter Strategy and Sensitivity**
>
> > **Reviewer:** The strategy to determine and evaluate the hyper-parameters is unclear. No ablation on different types of losses or sensitivity analysis to variations of α and β is provided.
>
> We appreciate the suggestion. During development, we performed an internal hyperparameter sweep for the spectral loss terms (**α** for KL divergence, **β** for cosine similarity) and the overall loss weights (**w₁–w₄**).
>
> We observed that the spectral loss weight (**w₃**) had the most notable impact. We have therefore included an ablation for **w₃** in **Appendix C.3** and **Table 6**, showing its influence on reconstruction quality. We found **w₃ = 0.2** to offer the best trade-off. We will revise Section 3.3 to reflect this tuning strategy and point to the new appendix section.
>
> ---
>
> ### **5. On Memory and Complexity Analysis**
>
> > **Reviewer:** An explicit discussion in terms of space (memory) overhead/savings is missing. No analysis on training complexity is provided.
>
> We thank the reviewer for this observation. In the revised paper (**Appendix C.4** and **Table 7**), we provide a detailed breakdown of training complexity, GPU memory usage, and inference speed across different variants of our model.
>
> ---

---

> > ### Author Response · Authors · 2025-07-03
> > **Part-2**
> >
> > ### **6. On Code Release**
> >
> > > **Reviewer:** Are the authors planning to release the code open-source?
> >
> > Yes—we **fully intend to release the code, pretrained models, and evaluation scripts** upon publication. We believe this will support reproducibility and allow the community to build on **HS-GS**. We will update the paper with a public GitHub repository link once the review process concludes.

---

### Review · Reviewer_G4a6 · 2025-06-20

**Summary Of Contributions:**

To address the problems of 3D reconstruction and novel view synthesis from hyperspectral images, a new method called **HS-GS** is proposed. It consists of three key technical modules:

- Wavelength Encoder: Learns to map each wavelength to an offset of Gaussian spherical harmonic coefficients in order to capture fine-grained spectral variations.

- Spectral Loss Function Based on Kullback–Leibler (KL) Divergence: Aligns the rendered results with the ground-truth spectra.

- Diffusion Denoising Module: Generates high-fidelity hyperspectral images.

**Audience:**

Yes

**Claims And Evidence:**

Yes

**Requested Changes:**

Please see the weakness, and these concerned are encouraged to be addressed.

**Strengths And Weaknesses:**

Strengths
- In multiple hyperspectral scenes, HS-GS significantly outperforms existing methods in metrics such as PSNR and SSIM, demonstrating the practical effectiveness of each module.

- The use of KL spectral loss is physically meaningful; this loss function encourages the rendered results to align with the true spectral distribution at each pixel, thereby improving spectral fidelity.

Weaknesses

-  **Lack of methodological innovation.**
The proposed method seems to loosely connect different modules and lack finely task-oriented design. Specifically, the proposed "Hyperspectral Gaussian Splatting" framework, though integrating components such as 3D Gaussian Splatting, a wavelength encoder, and a diffusion model, shows limited technical novelty as a whole. Specifically, the introduction of the diffusion model lacks task-driven consideration—it behavies more like a post-processing module typically used in image reconstruction tasks, rather than a systematic improvement derived from the nature of hyperspectral modeling. While diffusion models are widely applied in denoising and generation tasks for natural images, directly transferring them to 3D reconstruction of hyperspectral images—merely to improve visual metrics such as PSNR—lacks a theoretically or task-motivated rationale.

-  **Limited design of wavelength encoding, and insufficient exploration of Gaussian parameter modulation.**
  The paper proposes a wavelength-based offset strategy for spherical harmonic coefficients (SH offset), a design that has already been widely adopted in prior works on dynamic scene modeling and multispectral modeling. However, the authors do not explain why only the spherical harmonic terms are modulated, without considering the possibility of spectral modulation for other Gaussian kernel parameters (e.g., rotation matrix, scale matrix, opacity). Spectral-dependent modulation of different Gaussian properties has been explored in existing literature; a theoretical or empirical justification here would significantly enhance the persuasiveness of the approach.

-  **Lack of cost analysis and generalization evaluation for the diffusion model.**
   The diffusion model introduces considerable computational and memory overhead, yet the paper provides no systematic analysis of this cost—including training time, inference speed, or resource usage. Furthermore, it remains unclear whether applying the same diffusion module to outputs from vanilla 3DGS would yield similar performance gains.

-  **Limited and homogeneous experimental dataset.**
   Experiments are conducted only on five relatively simple and homogeneous plant scenes, lacking evaluation on more diverse datasets like ScanNetv200, as used in HyperGS.

---

> ### Author Response · Authors · 2025-07-03
> **Thanks for your review**
>
> “The introduction of the diffusion model lacks task-driven consideration—it behaves more like a post-processing module typically used in image reconstruction tasks, rather than a systematic improvement derived from the nature of hyperspectral modeling.”
>
> Author Rebuttal:
> We thank the reviewer for raising this concern. In our experiments, we evaluated autoencoders (please see Appendix C.4) to learn a latent representation for hyperspectral outputs. However, spectral distortions — particularly wavelength artifacts — were difficult to eliminate in a compressed latent space. These models struggled to preserve spectral continuity and channel-wise details due to their global encoding-decoding nature.
>
> Motivated by this limitation, we incorporated a diffusion model as a denoising mechanism that explicitly operates in the image pixel space and leverages the Markovian property of iterative refinement. This enables the model to learn a trajectory from noisy, artifact-prone renderings to high-fidelity hyperspectral outputs. Thus, while the diffusion module serves a denoising role, it is not arbitrarily appended, but introduced as a targeted solution to failure cases in spectral fidelity observed with standard approaches.

---

> > ### Author Response · Authors · 2025-07-03
> > **Part-2**
> >
> > “The authors do not explain why only the spherical harmonic terms are modulated, without considering the possibility of spectral modulation for other Gaussian kernel parameters (e.g., rotation matrix, scale matrix, opacity).”
> >
> > Author Rebuttal:
> > We appreciate the reviewer’s insightful comment. In hyperspectral imaging, object geometry remains constant across wavelengths—position, rotation, and scale do not change with light frequency. What varies is the material reflectance profile, corresponding to wavelength-dependent radiance.
> >
> > Therefore, we chose to modulate only the Spherical Harmonics (SH) coefficients of each 3D Gaussian, which encode view-dependent appearance rather than structural properties. This allows each Gaussian to emit radiance specific to each wavelength band while preserving shared geometry across channels.
> >
> > While prior works in dynamic or multispectral modeling modulate all parameters to capture physical or temporal scene changes, hyperspectral data assumes a static scene. Modulating geometric attributes in our case would introduce physically inconsistent deformations that violate the imaging assumptions.
> >
> > “Lack of cost analysis and generalization evaluation for the diffusion model. The diffusion model introduces considerable computational and memory overhead, yet the paper provides no systematic analysis of this cost—including training time, inference speed, or resource usage. Furthermore, it remains unclear whether applying the same diffusion module to outputs from vanilla 3DGS would yield similar performance gains.”
> >
> > Author Rebuttal:
> > We thank the reviewer for this helpful suggestion. In the revised Appendix C.3, we include a detailed cost breakdown comparing our method with and without the diffusion module in terms of training time, inference speed, GPU memory usage, and rendering FPS. As noted, diffusion introduces overhead (2.4 FPS vs. 78 FPS for vanilla 3DGS), but leads to significant reconstruction improvements.
> >
> > For generalization evaluation, we include a “3DGS + diffusion” configuration (no spectral components), which improves denoising and geometric fidelity over vanilla 3DGS. However, it lacks wavelength-aware supervision and spectral loss, and underperforms the full HS-GS pipeline. The spectral components are critical for channel-level accuracy and physical interpretability. Additional ablations are included in Table 3.
> >
> > “Limited and homogeneous experimental dataset. Experiments are conducted only on five relatively simple and homogeneous plant scenes, lacking evaluation on more diverse datasets like ScanNetv200, as used in HyperGS.”
> >
> > Author Rebuttal:
> > We thank the reviewer for pointing this out. While we agree that broader dataset diversity would be beneficial, it is important to highlight that no publicly available room-scale, multi-view hyperspectral dataset currently exists to our knowledge.
> >
> > Although prior work such as HyperGS uses a simulated dataset based on ScanNetV2 and RRUFF spectra, it is worth noting that the ScanNetV2 hyperspectral variant is not open-sourced. The dataset uses simulated spectra which is generated by mapping downsampled Raman spectra from the RRUFF database to segmentation labels. This does not reflect true hyperspectral imaging physics, spatial-spectral correlations, or sensor noise characteristics.
> >
> > We agree that extending evaluation to more diverse scenes is valuable and are actively exploring this avenue for future work.

---

> > > ### Author Response · Authors · 2025-07-04
> > > **Re-formatted Rebuttal**
> > >
> > > ---
> > >
> > > ### **On the Task-Driven Motivation for the Diffusion Model**
> > >
> > > > **Reviewer:** “The introduction of the diffusion model lacks task-driven consideration—it behaves more like a post-processing module typically used in image reconstruction tasks, rather than a systematic improvement derived from the nature of hyperspectral modeling.”
> > >
> > > **Author Rebuttal:** We thank the reviewer for raising this concern. In our experiments, we evaluated autoencoders (please see **Appendix C.4**) to learn a latent representation for hyperspectral outputs. However, spectral distortions — particularly wavelength artifacts — were difficult to eliminate in a compressed latent space. These models struggled to preserve spectral continuity and channel-wise details due to their global encoding-decoding nature.
> > >
> > > Motivated by this limitation, we incorporated a **diffusion model** as a **denoising mechanism** that explicitly operates in the image pixel space and leverages the Markovian property of **iterative refinement**. This enables the model to learn a trajectory from noisy, artifact-prone renderings to high-fidelity hyperspectral outputs. Thus, while the diffusion module serves a denoising role, it is not arbitrarily appended, but introduced as a **targeted solution** to failure cases in spectral fidelity observed with standard approaches.
> > >
> > > ---
> > > ## On Modulating Only Spherical Harmonics
> > >
> > > > “The authors do not explain why only the spherical harmonic terms are modulated, without considering the possibility of spectral modulation for other Gaussian kernel parameters (e.g., rotation matrix, scale matrix, opacity).”
> > >
> > > We appreciate the reviewer’s insightful comment. In hyperspectral imaging, an object's **geometry** (position, rotation, and scale) remains constant across wavelengths. What changes is the material's reflectance profile, which corresponds to wavelength-dependent radiance or **appearance**.
> > >
> > > Therefore, we chose to modulate only the **Spherical Harmonics (SH) coefficients** of each 3D Gaussian, as they encode this view-dependent appearance rather than structural properties. This allows each Gaussian to emit radiance specific to each wavelength band while preserving a single, shared geometry across all channels. Modulating geometric attributes would introduce physically inconsistent deformations, violating the core assumptions of hyperspectral imaging for static scenes.
> > >
> > > ---
> > >
> > > ## On Cost Analysis and Generalization
> > >
> > > > “Lack of cost analysis and generalization evaluation for the diffusion model. The diffusion model introduces considerable computational and memory overhead, yet the paper provides no systematic analysis of this cost...it remains unclear whether applying the same diffusion module to outputs from vanilla 3DGS would yield similar performance gains.”
> > >
> > > We thank the reviewer for this helpful suggestion. In the revised **Appendix C.3**, we include a detailed cost breakdown comparing our method with and without the diffusion model, covering training time, inference speed, GPU memory usage, and rendering FPS. As noted, diffusion introduces overhead (**2.4 FPS** vs. **78 FPS** for vanilla 3DGS) but leads to significant reconstruction improvements.
> > >
> > > For generalization, we added a **“3DGS + diffusion”** configuration to our ablations in **Table 3**. While this setup improves denoising over vanilla 3DGS, it underperforms the full **HS-GS** pipeline because it lacks wavelength-aware supervision and our spectral loss. This highlights that the spectral components are critical for achieving high channel-level accuracy and physical interpretability.
> > >
> > > ---
> > >
> > > ## On Experimental Dataset Diversity
> > >
> > > > “Limited and homogeneous experimental dataset. Experiments are conducted only on five relatively simple and homogeneous plant scenes, lacking evaluation on more diverse datasets like ScanNetv200, as used in HyperGS.”
> > >
> > > We thank the reviewer for pointing this out. While we agree that broader dataset diversity would be beneficial, it's important to highlight that no publicly available, room-scale, multi-view hyperspectral dataset currently exists to our knowledge.
> > >
> > > Although prior work like **HyperGS** uses a simulated dataset based on **ScanNetV2** and **RRUFF** spectra, that specific hyperspectral variant is not open-sourced. Furthermore, it uses simulated spectra generated by mapping Raman data to segmentation labels, which doesn't fully reflect true hyperspectral imaging physics, spatial-spectral correlations, or real-world sensor noise. We agree that extending evaluation to more diverse scenes is a valuable direction for future work and are actively exploring this avenue.

---

### Decision · Action_Editor_zC6x · 2025-08-04

**Recommendation:** Reject

**Audience:**

Yes

**Audience Explanation:**

The manuscript tackles a problem that is relevant for some practical applications, e.g. for non-destructive estimation of plant nutrient composition.

**Claims And Evidence:**

No

**Claims Explanation:**

The authors claim: "To enhance the model's ability to capture fine-grained reflectance variations across the light spectrum and leverage correlations between adjacent wavelengths for denoising, we introduce a wavelength encoder to generate wavelength-specific spherical harmonics offsets."

The reviewers were concerned about this point in several ways. First, they noted that the diffusion model is attributed to the main improvement while the contribution of the wavelength encoder is remains unclear. Second, it was noted that "leverage correlations between adjacent wavelengths for denoising" is at odds with the remark of the authors that their intention is not to enforce smoothness of a prior.

In addition, the modeling assumptions behind the forward diffusion process and their validity are unclear, which entails a potential technical problem of the manuscript.

**Resubmission Of Major Revision:**

The authors may consider submitting a major revision at a later time.